# Map It Anywhere (`MIA`): Empowering Bird's Eye View Mapping using Large-scale Public Data

**Cherie Ho**[1*]     **Jiaye Zou**[1*]     **Omar Alama**[1*]     **Sai Mitheran Jagadesh Kumar**[1]

**Benjamin Chiang**[1]     **Taneesh Gupta**[1]     **Chen Wang**[2]     **Nikhil Keetha**[1]

**Katia Sycara**[1]     **Sebastian Scherer**[1]

[1]**Carnegie Mellon University**     [2]**University at Buffalo**

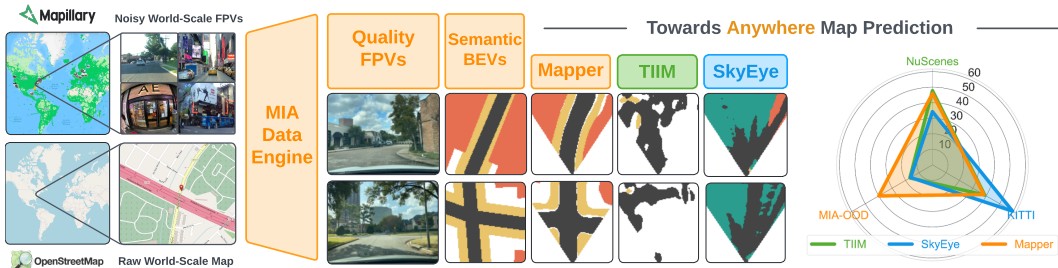

Figure 1: Our **M**ap **I**t **A**nywhere (`MIA`) data engine empowers *generalizable* Bird's Eye View (BEV) map prediction from First-Person View (FPV) images. *Left:* `MIA` enables seamless automatic curation of quality FPV & semantic BEV map data from *crowd-sourced* platforms, Mapillary & OpenStreetMap. *Right:* Both as a tool for training & benchmarking, `MIA` enables research towards *anywhere* map prediction. A simple model (`Mapper`) trained on data from `MIA` better generalizes on both held-out cities (`MIA-OOD`) & existing benchmarks, while state-of-the-art baselines trained on conventional autonomous vehicle datasets struggle.

## Abstract

Top-down Bird's Eye View (BEV) maps are a popular perceptual representation for ground robot navigation due to their richness and flexibility for downstream tasks. While recent methods have shown promise for predicting BEV maps from First-Person View (FPV) images, their generalizability is limited to small regions captured by current autonomous vehicle-based datasets. In this context, we show that a more scalable approach towards generalizable map prediction can be enabled by using two large-scale crowd-sourced mapping platforms, Mapillary for FPV images and OpenStreetMap for BEV semantic maps. We introduce **M**ap **I**t **A**nywhere (`MIA`), a data engine that enables seamless curation and modeling of labeled map prediction data from existing open-source map platforms. Using our `MIA` data engine, we display the ease of *automatically* collecting a dataset of *1.2 million* pairs of FPV images & BEV maps encompassing diverse geographies, landscapes, environmental factors, camera models & capture scenarios. We further train a simple camera model-agnostic model on this data for BEV map prediction. Extensive evaluations using established benchmarks and our dataset show that the data curated by `MIA` enables *effective pretraining for generalizable BEV map prediction*, with zero-shot performance far exceeding baselines trained on existing datasets by 35%. Our analysis highlights the promise of using large-scale public maps for developing & testing generalizable BEV perception, paving the way for more robust autonomous navigation. Website: mapitanywhere.github.io

---

*Equal contribution.

38th Conference on Neural Information Processing Systems (NeurIPS 2024) Track on Datasets and Benchmarks.

# 1   Introduction

Bird's Eye View (BEV) maps are an important perceptual representation for localization[20, 33], mapping [3, 15, 22, 29, 32], and decision-making tasks [18, 27], particularly for ground robots. They provide a rich, efficient metric representation of the world, enabling spatial reasoning directly in the plane ground robots move in. Given these advantages, autonomous driving systems often employ BEV maps as their primary perceptual representation [18, 42]. Beyond on-road driving, BEV mapping is widely used in other robot domains, such as offroad driving[1, 27, 36], mobile manipulation [40], and exploration [12, 16]. To enable wide deployment of BEV maps as a perceptual representation, there is a requirement for a *general map prediction building block* that performs robustly across different domains and supports effective adaptation to specific tasks/environments.

Despite tremendous advancements in predicting BEV maps from First Person View (FPV) images [15, 26, 32], we find that achieving good out-of-the-box predictions across diverse scenarios remains challenging. This shortcoming mainly stems from the current training & testing paradigm on limited-scale datasets collected using autonomous vehicle (AV) platforms [4, 5, 23, 35, 39]. While these benchmarks have massively propelled the field, they are *principally limited* in capturing large-scale diversity due to the time and cost associated with manual labeling, the limited deployment range, and finally, the use of specific sensor configurations on current AV stacks.

We believe that a complementary training & testing paradigm is necessary to assess the generalizability & robustness of BEV mapping, paving the way for *anywhere* deployment. Hence, starting from first principles, we formulate the key requirements for generalizable BEV mapping as: (a) being able to provide top-down information of key navigation classes, (b) ability to be used by different agents and across different operating regimes, for example, sidewalk prediction is more critical for autonomous wheelchairs, (c) perform reasonably out-of-the-box in unseen locations supporting quick adaptation, and (d) easily adaptable to different hardware configurations such as camera models.

In this context, we explore the question of **"How can one collect a dataset to empower generalizable BEV mapping?"** Specifically, to support research on generalizable BEV mapping, such a dataset needs to (a) contain diverse geographies, terrain types, time of day, and seasons, (b) capture scenarios beyond on-the-road driving, (c) support various camera models, and (d) consists of well-distributed classes and labels for supporting navigation.

To construct such a dataset, our key insight is to leverage two disjoint, crowd-sourced, and *world-scale* public mapping platforms: Mapillary for First-Person View (FPV) images and OpenStreetMap for Bird's Eye View (BEV) semantic maps. Both open-source platforms provide the tools necessary to associate crowd-sourced FPV images with semantic raster maps used for everyday human navigation. We introduce `MIA`, a data engine which taps into the potential of these mapping platforms to enable seamless curation and modeling of labeled data for generalizable BEV map prediction. Specifically, our data engine enables an evergrowing BEV dataset and benchmark, which exhibits world-scale diversity and supports research on both universal & environment-specific deployment.

In this paper, to showcase the potential of our data engine, we make the following key contributions:

1. We open-source our `MIA` data engine for supporting automation curation of paired world-scale FPV & BEV data, which can be readily used for BEV semantic map prediction research.

2. Using our `MIA` data engine, we release a dataset containing ∼1.2 million high quality FPV image and BEV map pairs covering 470 $km^2$, thereby facilitating future map prediction research on generalizability and robustness.

3. We show that training a simple camera intrinsics-agnostic model with our released datasets results in superior zero-shot performance over existing state-of-the-art baselines on key static classes, such as roads and sidewalks.

4. Through analysis of current performance in urban and rural domains of our benchmark, we show that significant research remains to enable generalizable BEV map prediction.

Overall, `MIA` establishes a diverse evergrowing dataset & benchmark for map prediction research and showcases how commodity public maps can empower generalizable BEV perception tasks (Fig. 1).

Table 1: **Statistics showcasing the broader scale of `MIA` in comparison to prior BEV Datasets.** Taxonomy: U: Urban, S: Suburban, R: Rural, O: Offroad, BN: Boston, SP: Singapore. "-": Attributes are not available. "*": MGL is a BEV localization dataset and does not provide semantic BEV maps suitable for map prediction. "# BEV Annotated Frames": Readily available BEV data. "Automatic Curation": No human intervention in the collection and annotation of the dataset.

| Dataset | Locations | km² covered | # Annotated BEV Frames | # Camera Models | Domain types | | | | Capture Platform | | | Suitable for Map Pred. | Automatic Curation |
|---|---|---|---|---|---|---|---|---|---|---|---|---|---|
| | | | | | U | S | R | O | Car | Bik | Ped | | |
| Argoverse [5] | 2 in US | 1.6 [35] | 22K | 2 | ✓ | X | X | X | ✓ | X | X | ✓ | X |
| Argoverse 2 [39] | 6 in US | - | ∼108K [43] | 2 | ✓ | X | X | X | ✓ | X | X | ✓ | X |
| KITTI-360-BEV [15] | 1 in DE | 5.3 | 83K | 2 | X | ✓ | ✓ | X | ✓ | X | X | ✓ | X |
| NuScenes [4] | BN, SP | 5.6 | 40K | 2 | ✓ | ✓ | X | X | ✓ | X | X | ✓ | X |
| Waymo [35] | 3 in US | 76 [35] | 230K | 2 | ✓ | ✓ | X | X | ✓ | X | X | ✓ | X |
| Orienternet (MGL)* [33] | 12 in US/EU | - | 760K | 4 | ✓ | ✓ | ✓ | X | ✓ | X | X | X | X |
| **`MIA` (Ours)** | 6 in US | 470 | 1.2M | 17 | ✓ | ✓ | ✓ | ✓ | ✓ | ✓ | ✓ | ✓ | ✓ |

## 2 Related Work

**Bird's Eye View Map Prediction:** BEV map prediction involves predicting top-down semantic maps from various sensory modalities to facilitate downstream robotic tasks. Some works rely solely on LiDAR [21], others on multi-view cameras [30], and some on both [24, 46]. These approaches rely on modalities that are expensive and difficult to calibrate. Recently, a growing number of works use *monocular* cameras [14, 15, 25, 26, 31, 32], as they are attractive for their ease of deployment, reduced cost, and higher scalability. However, all these works still rely on current autonomous driving datasets for labels, which limits the scalability of data collection. To address this limitation, SkyEye [15] uses more available front-view semantics to build map predictors without explicit BEV maps. However, this method relies on ground truth FPV semantic masks, which are costly to annotate and scale. In contrast, `MIA` leverages two readily available world-scale databases to provide diverse and accurate supervision, avoiding the high cost of equipment and manual labor.

Another line of related work is the task of matching FPV images with BEV maps that can be satellite-based [17, 45], planimetric [33, 41], or multi-modal [34]. They often employ techniques to predict a BEV feature map given an FPV image. While useful for retrieval or localization, these feature maps cannot benefit downstream tasks without complex learned decoders, unlike predicted semantic BEV maps, which downstream algorithms like path planning can readily consume.

**Datasets for BEV Map Prediction:** Existing BEV map prediction datasets are often derived from multi-modal autonomous driving datasets [4, 5, 14, 15, 35, 39] that target various tasks, including BEV prediction. These pioneering works depend on manually collected data from costly LiDARs, hence requiring careful calibration with camera setups to ensure accurate correspondence between FPV & BEV data. The BEV is generated by accumulating semantically labeled LiDAR point clouds & then splatting them to BEV, allowing them to capture dynamic and static classes. However, these approaches are **principally limited** in both **scale** and **diversity** due to their high cost, hindering model generalizability. In contrast, `MIA` uses data available on crowd-sourced platforms and can thus obtain FPV-BEV pairs globally, achieving broader diversity & scale as shown in Table 1.

**Crowd-sourced Datasets for Learning Geometric Tasks:** Crowd-sourced platforms enable open-source contributors to upload diverse in-the-wild data, significantly empowering generalizability in geometric learning tasks. One such notable platform is Mapillary [7], which hosts over 2 billion (and growing) crowd-sourced street-level images from various locations worldwide, captured by different cameras across all seasons and times. Mapillary has been notably used for tasks such as depth estimation [2], lifelong place recognition [38], and visual localization [19, 33]. Most related to our work, OrienterNet [33] addresses visual localization within a large top-down map, curating a large-scale localization dataset provided by crowd-sourced platforms, Mapillary [7] for FPV images and OpenStreetMap [9] (OSM) for large BEV maps. However, the OrienterNet pipeline for rasterizing maps is not suitable for BEV prediction as it renders large maps, mimics OSM style, and includes graphical elements/labels irrelevant to the map prediction task. We illustrate the qualitative differences in Figs. 11 and 12 (shared in the Appendix). `MIA` further refines the OrienterNet pipeline by enabling automatic curation and collection of semantic maps, alignment of BEV renders with satellite images, and inference of missing OSM sidewalk geometries, thus providing rich semantic maps that are ready for BEV map prediction.

# 3 `MIA` Data Engine

To construct a dataset for generalizable BEV mapping, we develop a scalable data engine that generates high-quality, diverse FPV-BEV pairs with rich semantic labels. This process, summarized in Fig. 2 and detailed below, follows the criteria discussed in Section 1.

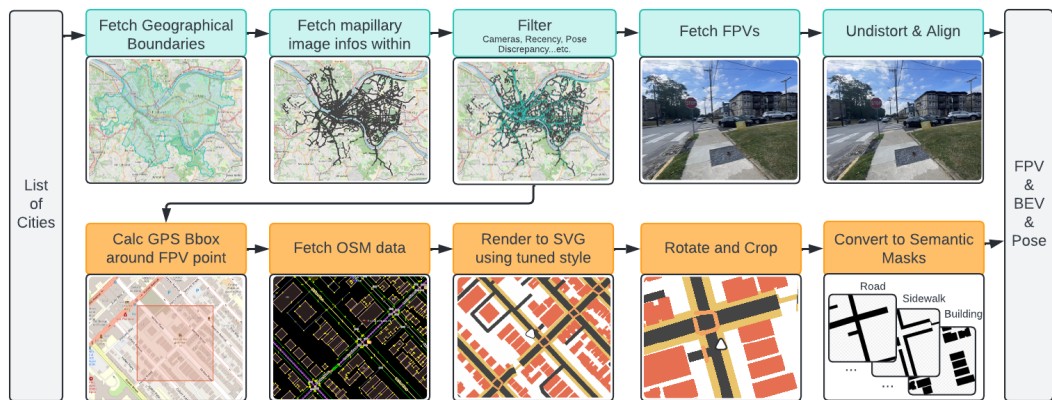

Figure 2: **Overview of how the `MIA` data engine enables automatic curation of FPV & BEV data.** Given names of cities as input from the left, the top row shows FPV processing, while the bottom row depicts BEV processing. Both pipelines converge on the right, producing FPV, BEV, and pose tuples.

## 3.1 First Person View (FPV) Retrieval

**Mapillary:** For FPV retrieval, we leverage Mapillary [7], a massive public database, licensed under *CC BY-SA*, with over 2 billion crowd sourced images. The images span various weather and lighting conditions collected using diverse camera models and focal lengths. Furthermore, images are taken by pedestrians, vehicles, bicyclists, etc. This diversity enables the collection of more dynamic and difficult scenarios critical for *anywhere* map prediction. However, this massive pool of data is not readily amenable to deep learning as it contains many noisy, distorted, and incorrectly registered instances. Thus far, few works, such as [33], have leveraged such data providing an impressive retrieval and undistortion pipeline. However, the work relied on careful and manual curation of limited camera models and scenarios. Such approaches are not scalable and cannot leverage the powerful quantity and diversity of Mapillary. Hence, we further refine the OrienterNet [33] data curation framework and develop a fully automated curation pipeline that can harness the full potential of the extensive Mapillary database. We describe the pipeline in the following sections.

**FPV Pipeline:** As demonstrated in Fig. 2, the FPV pipeline starts by manually inputting a list of locations of interest, which can be as simple as inputting the name of the location or as specific as specifying the GPS bounds. The geographical bounds are then fetched using the Nominatim API [8] if needed. The pipeline then converts these bounds to a list of zoom-14 tiles and uses the public Mapillary APIs to query for the image instances within these tiles. Only the retrieved instances that lie within the geographical boundaries of interest are kept. Given the retrieved image IDs, we use another Mapillary endpoint to retrieve image metadata, which includes coordinates rectified through structure from motion, camera information, poses, and timestamps, amongst other details that we use for filtering in the subsequent stage. For more information on the raw data used from Mapillary, please refer to Supplemental Information Q8.

We develop the filtering pipeline by observing hundreds of FPV & BEV pairs and identifying the correlations between good-quality FPVs and their corresponding metadata. The criteria we used included a recency filter, a camera model filter spanning 19 camera models with good RGB quality, a location/angle discrepancy filter that computes the difference between Structure from Motion (SfM) computed and recorded poses, both obtained from Mapillary API, as a proxy for measuring the quality of the geo-registration, and a camera type filter that only includes perspective and fisheye. To promote spatial diversity over sheer quantity, we filter out images within a 4-meter radius of another image from the same sequence. After filtering, we retrieve the RGB images from Mapillary

and feed them through an undistortion pipeline adapted from [33]. The undistortion is critical for fisheye images to ensure their pixel-aligned features can be correctly lifted into BEV space. Using this pipeline, we can retrieve high-quality images from anywhere in the world, tapping into the power of the Mapillary platform.

## 3.2 Birds Eye View (BEV) Retrieval

**Open Street Map (OSM):** For BEV retrieval, we leverage OSM [9], a global crowd-sourced mapping platform open-sourced under Open Data Commons Open Database License (ODbL). OSM provides rich vectorized annotations for streets, sidewalks, buildings, etc. However, OSM data cannot be easily used for map prediction as (a) OSM often does not encode critical sidewalk geometry, (b) structured formats like OSM maps prove difficult to train on, (c) off-the-shelf rendering pipelines target human consumption often encoding information irrelevant for BEV prediction (such as textual labels) and do not care about pixel aligning maps with satellite imagery, thereby encoding inaccurate road widths in many instances. Recognizing the need to create our own rasterization pipeline, we build on top of the MIT-licensed MapMachine project [37] and study hundreds of satellite/map pairs to achieve rasterization that is more pixel-aligned with satellite imagery. We further carefully map the hundreds of elements in OSM to a handful of informative dominant semantic labels.

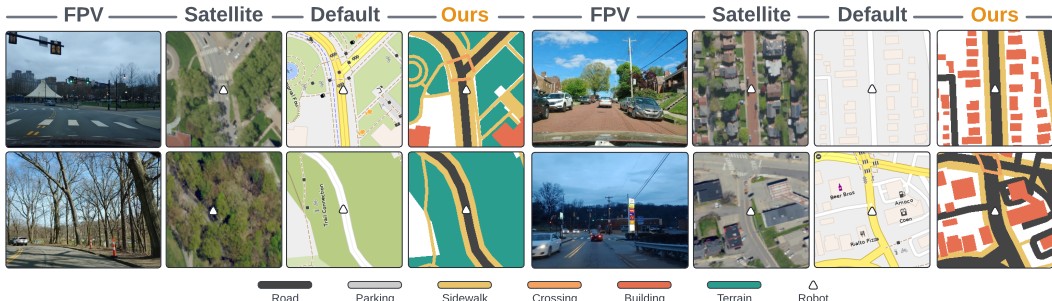

Figure 3: **Comparison of default MapMachine-style rendering with the `MIA`-style**. The figure shows our rendering removes irrelevant information, clusters key semantic categories, aligns better with satellite and is able to provide more accurate sidewalk geometry correctly. Satellite imagery is not part of the `MIA` data engine and was obtained from [13] only for tuning map rendering.

**BEV Generation Pipeline:** BEV retrieval starts after the filtering stage in FPV retrieval as illustrated in Fig. 2. Given coordinates in the World Geodetic System (WGS-84) frame for each image, we project each point onto a Cartesian UTM coordinate frame, estimated separately for each city/location. Next, we calculate an ego-centric bounding box of size $(\alpha + \beta + \delta)^2$ at a resolution of $\rho$ meters per pixel. Here, $\alpha$ represents the requested image dimension, $\beta = \alpha - \frac{\alpha}{\cos(\frac{\pi}{4})}$ is the padding added to accommodate rotations without introducing empty space, and $\delta$ is the padding added to avoid missing any OSM elements that may not fall within the original box. To adhere to the OSM API, we project boxes back to WGS-84 coordinates before retrieving OSM data for every image. We then utilize our version of MapMachine (enhanced to infer missing sidewalks from OSM) with a carefully tuned, satellite-aligned map style to render the data into SVG format. Next, we rotate the rendered image so that the robot is looking 'up' in the BEV image plane, aligning it with the forward direction of the FPV plane. Finally, we rasterize the SVG into a semantic mask containing six static classes (Road, Parking, Sidewalk, Crossing, Building & Terrain), as shown in Fig. 3, to produce the final BEV.

## 4 Empowering Map Prediction with the `MIA` Data Engine

### 4.1 Sampling the `MIA` Dataset

We show the utility of the `MIA` data engine by sampling six different urban-centered locations, extending to the suburbs. We selected highly populated cities - New York, Chicago, Houston, and Los Angeles - to collect challenging scenarios with diverse and dense traffic. Additionally, we included Pittsburgh and San Francisco for their unique topologies. For BEV retrieval, we set $\alpha = 224$, $\delta = 50$, and $\rho = 0.5$, resulting in what we believe is the largest public BEV prediction dataset, comprising

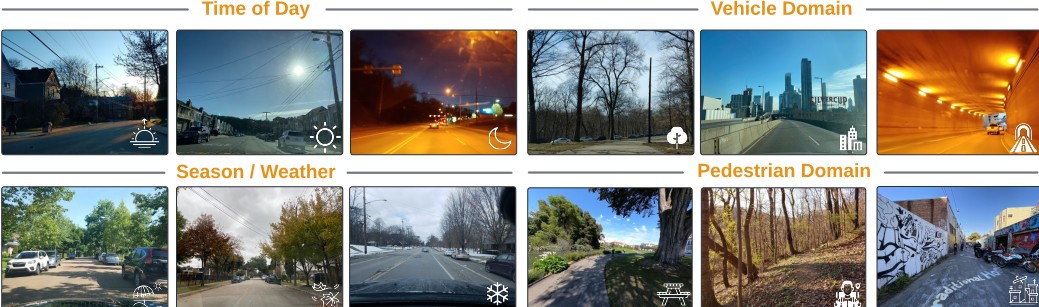

Figure 4: **Samples from the `MIA` dataset**: Highlighting diversity in time of day, seasons, weather and capture scenarios from vehicles & pedestrians.

approximately 1.2 million FPV-BEV pairs, as shown in Table 1. To illustrate the diversity of the data, we adapt the coverage metric proposed by [35] in which each image instance covers a radius of 150 meters around its pose. For our sampled dataset, we calculate the coverage at a radius of 112 meters consistent with our chosen $\alpha$ value. As shown in Table 1, our sampled dataset covers $470\ km^2$, far surpassing all existing BEV prediction datasets by $6\times$. This highlights the immense potential of our *scalable* `MIA` data engine to produce large quantities of annotated FPV-BEV pairs covering extensive geographies and with varying camera models and focal lengths, as highlighted in Fig. 4.

To further benchmark the generalization capability of map prediction models in more extreme settings, we further sample a small ($\sim 1.1K$) rural/remote dataset, which we denote as `MIA-Rural`. This dataset has a distribution very different from the urban-centered samples. We selected locations with distinct visual appearances, namely Willow (Alaska), Ely (Nevada), and Owen Springs (Australia). To push the boundaries of generalization testing, we disabled the camera model filter for this test set, thereby incorporating a variety of challenging camera models into this extreme dataset.

### 4.2 `Mapper`: **Training a camera intrinsics-agnostic baseline model**

We train a model, `Mapper`, with `MIA` to validate the need for such a large-scale and diverse dataset by testing its generalization capability. Leveraging the diversity of the Mapillary dataset requires a model architecture capable of handling various image characteristics, such as focal lengths and image size. Additionally, following OrienterNet [33], it is reasonable to assume that the robot or phone has orientation information (IMU), and we aim to incorporate this information into our map predictions.

Our goal is to learn a model that takes in a monocular image $\mathbf{I}$ to predict a gravity-aligned BEV semantic map $\mathbf{Y}$. Formally, given an image $\mathbf{I} \in \mathbb{R}^{3 \times H \times W}$, its intrinsic matrix $\mathbf{C} \in \mathbb{R}^{3 \times 3}$, and its extrinsic matrix $\mathbf{E} \in \mathbb{R}^{3 \times 4}$, we seek to produce a multi-label binary semantic map in gravity-aligned frame $\mathbf{Y} \in \mathbb{R}^{X \times Z \times K}$ where $K$ is the number of semantic classes. To achieve this, we build on OrienterNet [33], designed for top-down map localization, as the front-end architecture. This choice accommodates different camera characteristics and pose information from IMUs, thereby leveraging the orientation data from OSM and Mapillary. We add a decoder head on the BEV features (obtained post 2D-to-3D lifting of FPV features) to predict a semantic map, thereby maintaining simplicity in the model architecture. Furthermore, to improve generalization capability, we replace the ResNet encoder with the DINOv2 encoder [28]. For training, we resize and pad images to a 512 x 512 square, applying weighted Dice and Binary Cross Entropy Loss to the BEV pixels within the image frustum. We use DINOv2 ViT-B/14 [28] with registers [10] as the image encoder. We further augment the dataset with brightness, contrast, and color jittering. We train with a batch size of 128 for 15 epochs, which takes approximately 4 hours using 4 NVIDIA-H100 GPUs. The supplementary material provides more details on the model and training, along with a figure of the model pipeline.

## 5 Experimental Setup

To evaluate generalizability, we test our `Mapper` model and baselines on multiple datasets, including our diverse `MIA` dataset and conventional map prediction datasets.

**Conventional Datasets:** We evaluate our model on BEV map segmentation benchmarks: NuScenes [4] and KITTI360-BEV [15], to demonstrate the generalizability of a model trained on `MIA` when applied to established datasets. We follow the BEV generation procedure in [31] for NuScenes. Both datasets are collected from on-road vehicles and present challenges such as occlusions, different times of day, and varying weather conditions. We adhere to the conventional(geographically non-overlapping) data set splits: Roddick et al.'s [31] split for NuScenes and Gosala et al.'s split [15] for KITTI360-BEV. Since we focus on static class map prediction, we exclude dynamic elements from the dataset labels before training. For NuScenes, we use only the static map layers, and for KITTI360-BEV, we remove dynamic object labels. In the experiments, we use the front-facing camera data to evaluate monocular camera BEV map prediction. Class mappings between the datasets are provided in Table 6 of the appendix.

`MIA` **Dataset:** We utilize the `MIA` dataset to assess performance in diverse urban and rural settings not captured by previous datasets, including held-out environments. Specifically, we split our dataset into two settings: `MIA-ID` (New York, Los Angeles, San Francisco, Chicago) represents in-distribution urban areas, `MIA-OOD` (Houston, Pittsburgh) tests the generalization ability in held-out urban settings. Furthermore, we use `MIA-Rural` to provide a more challenging out-of-distribution evaluation. For each location, we generate an 80% train / 10% validation / 10% test split, ensuring the splits are geographically disparate as illustrated in Fig. 10 of the appendix.

**Metrics:** We adhere to standard conventions [15, 32] to calculate the Intersection-over-Union (IoU) score using binarized predictions with a threshold of 0.5. The IoU score is computed over the observable area as defined by the visibility mask, based on LiDAR observations or the visibility frustum. Since there is no LiDAR sensing in the `MIA` dataset, we generate a heuristic-based visibility mask by raycasting from the robot's position, ending 4 pixels into a building. To be consistent with SkyEye evaluations [15], in KITTI360-BEV, we also include occluded areas within image frustum for IoU calculation. For all datasets, we performed comparisons over a 50m x 50m area with a resolution of 0.5m/pixel. Consistent with prior work [15, 32], we report the macro-mean IoU over all classes. In addition, for a fair comparison across different methods and datasets, we report the macro-mean IoU over the two classes common in all datasets, *road & sidewalk*.

**Baselines:** We compare our results with previously published methods that focus on the monocular single-camera setting, specifically Translating Images into Maps (TIIM) [32], which was trained and tested on NuScenes [4], and SkyEye, which was tested on KITTI360-BEV [15]. While newer methods have been proposed [6, 44], TIIM is the most recent method with available code, to our knowledge. We follow the training protocols described in the respective papers and code, with slight modifications to train for static classes. For evaluation, images are processed to meet the requirements of each method. To test the baseline models on datasets they were not trained on, we follow Gosala et al. [15] and resize the image to match the focal length of the model's training dataset. More details on baseline implementation are provided in supplementary material and appendix.

## 6 Results & Discussion

We firstly evaluate `Mapper` zero-shot against the baselines. Next, we test `MIA`'s effectiveness for pre-training by finetuning `Mapper` on limited data from an existing dataset. Finally, we stress test all models in extreme out-of-distribution scenarios to highlight future opportunities enabled by `MIA`.

### 6.1 `MIA` can go more "anywhere" out-of-the-box

Table 2 demonstrates the generalizability of `Mapper`, trained with the `MIA-ID` dataset, over both zero-shot & fully-supervised baselines, in particular on the NuScenes and `MIA-OOD` environments. Fig. 5 visually compares the model predictions, where `Mapper` provides more realistic predictions across the datasets compared to the zero-shot baseline, which often fails due to the distribution change caused by unseen location, different camera models, or severe weather conditions. When comparing average IoU of Road and Sidewalk, `Mapper` achieves superior zero-shot result in NuScenes *val* [4] and `MIA-OOD`, with improvements of 33% and 144%, respectively. Notably, in NuScenes and `MIA-OOD`, `Mapper` performs comparably to fully supervised methods (trained with in-domain data) in road and sidewalk classes. While `Mapper` performs consistently with *TIIM* (trained on NuScenes) when tested on KITTI360-BEV [15], `Mapper` provides more realistic predictions overall, especially in non-road regions where TIIM tends to overpredict roads. However, due to the limitations of

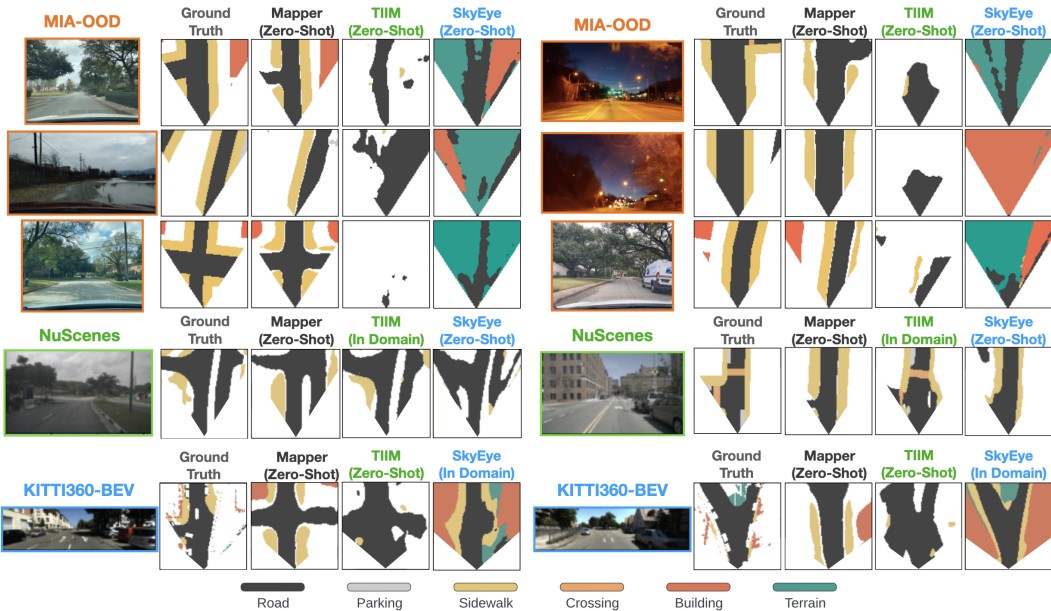

Figure 5: `Mapper` **consistently provides more precise & realistic zero-shot predictions across all the datasets.** Notably, `Mapper`, empowered by `MIA` data, can produce zero-shot predictions which are comparable to the fully-supervised baselines which have been trained on in-domain data.

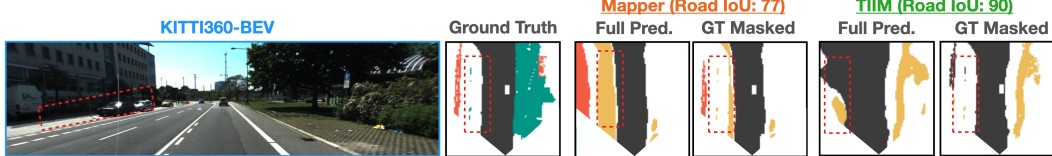

Figure 6: **Lack of complete labels in KITTI360-BEV [23] dilutes quality of benchmarking.** For example, while `Mapper` predicts sidewalk and road reasonably in this frame, the lack of sidewalk labels in the ground truth results in a misrepresentative IoU. Meanwhile, TIIM's [32] road IoU is artificially higher, despite the incorrect road prediction on the left.

KITTI360-BEV, where much of the map is unlabeled and IoU is only measured in labeled regions, as illustrated in Fig. 6, it is challenging to perform effective benchmarking and comparisons.

## 6.2   Pre-training on `MIA` enables effective fine-tuning with limited data

We also test if the `MIA` dataset can provide effective pretraining for new map prediction tasks. Specifically, we finetune `Mapper` with 10% and 1% of NuScenes data. For comparison, we use the same data split to train TIIM [32]. To fine-tune `Mapper`, we map the new training dataset classes to `MIA` classes as described in Table 6 of appendix. Details on training, fine-tuning and data splits are available in the appendix and supplementary material. Table 2d shows that `Mapper` can be effectively fine-tuned on specific environments with limited new map prediction data. Notably, the experiment suggests that `MIA` provides effective pretraining for map prediction, as fine-tuning the pre-trained `Mapper` yields improved results compared to training TIIM solely on the data subset.

## 6.3   `MIA` provides challenging settings for future work on anywhere map prediction

To test model generalizability, we further curate `MIA-Rural`, which is far from the training distribution. Fig. 7 shows an example predictions from highway images where all models, including our proposed `Mapper`, fail to generalize. Quantitatively, on the entirety of `MIA-Rural`, the average IoU between road and sidewalks (Avg. R, S) is 21.04 for `Mapper`, 20.55 for TIIM [32] and 18.62 for SkyEye [15]. This further illustrates the research need for an *anywhere* map prediction dataset.

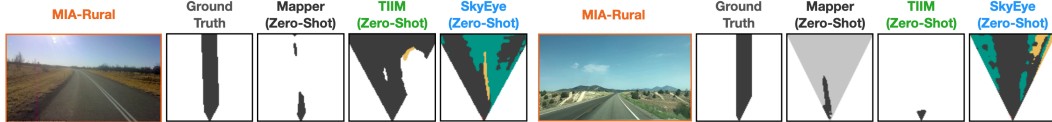

Figure 7: **Challenging scenarios mined using the `MIA` data engine:** We curate highway images far from urban environments to stress test models. This showcases our ability to extract challenging and high-impact test scenarios where current models, including `Mapper`, do not perform well.

Table 2: Benchmarking across all the datasets in both zero-shot & finetuning setups.

| Methods | In-Domain | Road | Crossing | Sidewalk | Carpark | Avg. | Avg. R, S |
|---|---|---|---|---|---|---|---|
| TIIM [32] | ✓ | 68.63 | 29.41 | 27.03 | 7.70 | 33.19 | 47.83 |
| SkyEye [15] | × | 52.57 | 0.00 | 15.47 | 0.00 | 17.01 | 34.02 |
| Mapper | × | **64.22** | **0.06** | **27.71** | 0.04 | **23.01** | **45.97** |

(a) Zero-Shot NuScenes [4]

| Methods | In-Domain | Road | Sidewalk | Building | Terrain | Avg. | Avg. R, S |
|---|---|---|---|---|---|---|---|
| SkyEye [15] | ✓ | 76.59 | 40.21 | 32.47 | 44.22 | 48.37 | 58.40 |
| TIIM [32] | × | **67.18** | 10.41 | 0.0 | 0.0 | 19.40 | **38.80** |
| Mapper | × | 58.92 | **11.99** | **25.08** | **0.60** | **24.15** | 35.46 |

(b) Zero-Shot KITTI360-BEV [15]

| Methods | In-Domain | Road | Crossing | Sidewalks | Building | Parking | Terrain | Avg. | Avg. R, S |
|---|---|---|---|---|---|---|---|---|---|
| Mapper+OOD | ✓ | 58.28 | 0.05 | 30.75 | 13.80 | 13.79 | 6.70 | 20.56 | 44.52 |
| TIIM [32] | × | 32.74 | 0.00 | 1.47 | 0.00 | 0.00 | 5.70 | 17.11 | 17.11 |
| SkyEye [15] | × | 33.09 | 0.00 | 1.18 | 5.70 | 0.00 | **2.67** | 7.11 | 17.14 |
| Mapper | × | **54.65** | **0.06** | **27.55** | **15.78** | 2.66 | 1.83 | **17.09** | **41.10** |

(c) Zero-Shot `MIA-OOD`

| Methods | Data % | Drivable | Crossing | Walkway | Carpark | Mean |
|---|---|---|---|---|---|---|
| Mapper | 0% | 61.30 | 0.51 | 25.50 | 0.00 | 21.83 |
| TIIM [32] | 1% | 57.80 | 3.11 | 12.66 | 0.10 | 18.42 |
| Mapper | 1% | 69.70 | 0.20 | 26.44 | 1.83 | 24.54 |
| TIIM [32] | 10% | 61.12 | 23.69 | 14.52 | 1.53 | 25.22 |
| Mapper | 10% | 73.10 | 10.80 | 19.21 | 5.37 | 27.12 |
| TIIM [32] | 100% | 68.63 | 29.41 | 27.03 | 7.70 | 33.19 |

(d) Finetuning `Mapper` on NuScenes [4] data

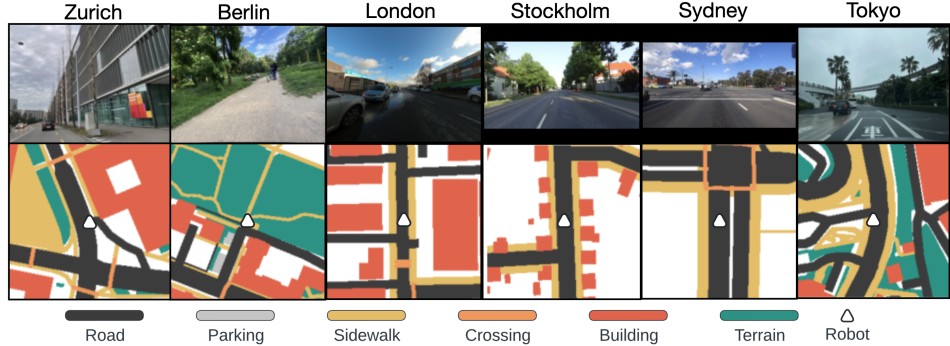

Figure 8: Example pairs of First-Person-View images & Birds-Eye-View semantic maps **effortlessly extracted** by our `MIA` data engine from non-US cities.

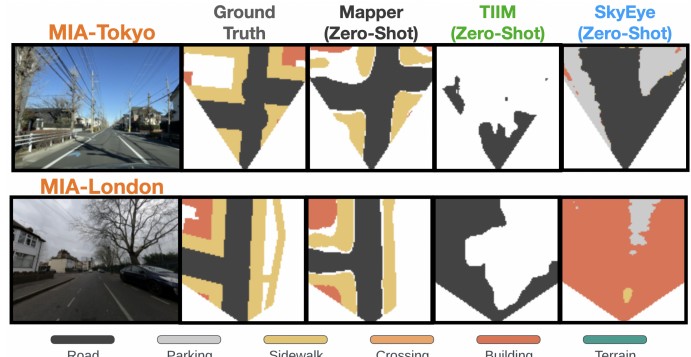

| Methods | Avg. over Classes | Avg. R, S |
|---|---|---|
| TIIM [32] | 10.78 | 21.21 |
| SkyEye [15] | 13.77 | 20.87 |
| Mapper | **17.66** | **37.26** |

Table 3: Zero-shot IoU results on `MIA-non-US` data split.

Figure 9: Zero-shot predictions in non-US cities. `Mapper` consistently provides more accurate & realistic zero-shot predictions.

## 6.4 `MIA` data engine can also effortlessly extract data beyond U.S. cities

To show that our `MIA` data engine can effortlessly curate data from other global locations, we curate `MIA-non-US` from 6 non-US cities spanning 3 continents: Europe (London, Stockholm, Berlin,

Zurich), Asia (Tokyo), and Australia (Sydney). Fig. 8 shows example data pairs. Furthermore, we leveraged this data to evaluate the generalizability of existing methods and `Mapper`. Results in Tab. 3 and Fig. 9 show that our model `Mapper` consistently provides more accurate and realistic zero-shot predictions compared to baselines trained on conventional datasets.

## 7 Conclusion

In this work, we propose `MIA`, a data curation pipeline aimed at empowering *anywhere* BEV map prediction from FPV images. We release a large `MIA` dataset obtained through the data engine giving the research community access to ~1.2M FPV-BEV pairs to accelerate *anywhere* map prediction research. Results from training on the dataset show impressive generalization performance across conventional map prediction datasets, while on the other hand, provides challenging test cases for the research community. Our approach departs from the traditional and expensive autonomous vehicle data collection and labeling paradigm, towards *automatic curation of readily-available crowd-sourced data*. We believe this work seeds the first step towards *anywhere* map prediction.

### 7.1 Limitations, Biases, Social Impact

While we show promising generalization performance on conventional datasets, we note that label noise inherently exists, to a higher degree than manually collected data, in crowd-sourced data, in both pose correspondence and in BEV map labeling. Such noise is common across large-scale automatically scraped/curated benchmarks such as ImageNet [11]. Moreover, our approach does not capture dynamic classes as they do not exist in static maps. However, we see our approach as indispensable for scale and diversity yet complementary to conventionally obtained datasets.

**Negative Societal Impact:** Our work relies heavily on crowd-sourced data, which places the burden of data collection on open-source contributors. Additionally, while the FPV images and metadata from Mapillary are desensitized, there remains a potential risk of reconstructing private information.

## Acknowledgments and Disclosure of Funding

The work is funded by National Institute of Advanced Industrial Science and Technology (AIST), Army Research Lab (ARL) awards W911NF2320007 and W911NF1820218 and W911NF20S0005, Defence Science and Technology Agency (DSTA). Omar Alama is partially funded by King Abdulaziz University. This work used Bridges-2 at PSC through allocation *cis220039p* from the Advanced Cyberinfrastructure Coordination Ecosystem: Services & Support (ACCESS) program which is supported by NSF grants #2138259, #2138286, #2138307, #2137603, and #213296.

We thank Mihir Sharma for his work on related research curation and exploration into potential extensions involving terrain height extraction. We thank Mononito Goswami for invaluable discussions on paper organization, framing, experimental design, and project feedback. Additionally, we appreciate David Fan, Simon Stepputtis, and Yaqi Xie for their insightful feedback throughout the project. We are grateful to our reviewers for their constructive feedback.

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

# A  Appendix

## A.1  Filtering Pipeline Yields

Table 4 shows the yield of data samples from the different stages of the `MIA` curation pipeline. Table 5 presents a statistics breakdown on the locations.

Table 4: The FPV Filtering pipeline stage for the `MIA` dataset, covering 6 cities, starts by filtering images based on city boundaries and recency (images after 2017). It then selects 17 camera models and filters images based on location and angle discrepancies between the SfM rectified pose and recorded pose (keeping those with less than $20°$ and $3m$ discrepancies). Finally, a spatial sparsity filter removes images within a $4m$ radius in a sequence.

| Curation Stage | Boundaries | Recency | Camera Model | Angle Discrip | Loc Discrip | Spatial |
|---|---|---|---|---|---|---|
| # Images | 15.93M | 12.21M | 3.049M | 2.606M | 1.353M | **1.204M** |
| % Images | 100.00% | 76.67% | 19.15% | 16.36% | 8.50% | **7.56%** |

Table 5: FPV numbers for the `MIA` Dataset as the curated data moves from the beginning of the pipeline (top of table) down to the end. PT: Pittsburgh, NY: New York, CG: Chicago, LA: Los Angeles, SF: San Francisco, HS: Houston.

| Stage | PT | NY | CG | LA | SF | HS | ALL |
|---|---|---|---|---|---|---|---|
| Boundaries | 914.1K | 3.161M | 1.422M | 4.150M | 2.825M | 3451961 | 15.93M |
| Recency | 867.0K | 2.999M | 1.281M | 4.055M | 2.100M | 908.5K | 12.21M |
| Camera Model | 31.0K | 270.5K | 478.5K | 1.917M | 294.9K | 57.3K | 3.049M |
| Angle Discrip | 25.7K | 177.6K | 433.6K | 1.758M | 156.9K | 54.0K | 2.606M |
| Loc Discrip | 18.0K | 89.8K | 196.6K | 958.6K | 59.2K | 31.1K | 1.353M |
| Spatial | 15.9K | 79.2K | 162.4K | 879.4K | 37.7K | 29.4K | **1.204M** |

## A.2  Dataset Split Visualization

`MIA` dataset is split into train, validation, and test partitions based on the location. We ensure that the samples in the three partitions are geographically non-overlapping, as shown in Fig. 10.

## A.3  Intra-Dataset Class Mappings

As different dataset captures different map elements, we map the classes from NuScenes and KITTI360-BEV carefully to `MIA`, as shown in Table 6.

Table 6: Intra-Dataset Class Mapping for Zero-Shot Experiments

| Mapper | NuScenes [32] | KITTI360-BEV [15] |
|---|---|---|
| Road | Drivable | Road |
| Crossing | Crossing | N/A |
| Sidewalks | Walkway | Sidewalk |
| Building | N/A | Building |
| Parking | Carpark | N/A |
| Terrain | N/A | Terrain |

## A.4  Additional Comparisons between MapItAnywhere and OrienterNet [33]

`MIA` builds on OrienterNet [33] which uses Mapillary [7] and OpenStreetMap [9], to perform RGB image localization within a pre-existing map. Our work extends OrienterNet [33] by using the same data sources to achieve the goal of generalizable FPV-to-BEV *map prediction*. Our work differs in 2 ways: (1) dataset processing pipeline and (2) released datasets. First, OrienterNet has a fixed set

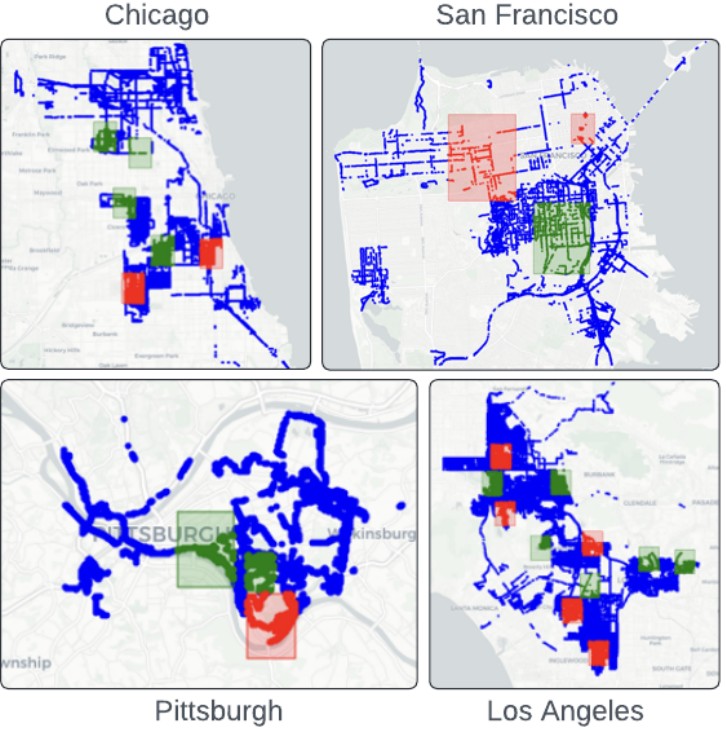

Figure 10: `MIA` **Dataset split visualization.** Blue for training, green for validation, and red for testing. We ensure that the splits are geographically non-overlapping for a more robust evaluation.

of image IDs that was curated either manually or through a closed-source method. In contrast, we developed a filtering pipeline that allows the automatic curation and collection of images from any location in the world. Users can define a city and change filtering thresholds to meet their needs. Secondly, while OrienterNet-rendered maps provide useful context to an image encoder, they are unsuitable as a prediction target for BEV prediction. This is because the map elements are not consistent with the real world / FPV image. E.g., roads in OrienterNet are 1 pixel wide, whereas the road in ours is a rasterized shape aligned with satellite and FPV images. We illustrate the differences in Figures 11 and 12.

### A.5 Additional `MIA` data samples

Additional data samples from `MIA` dataset and associated Mapillary metadata are shown in Figure 13.

### A.6 `Mapper` Model Architecture

We design a simple model architecture `Mapper` that can leverage the diverse characteristics of the `MIA` dataset. As discussed in the main paper, our model builds on OrienterNet [33] as the front-end architecture. This choice is made because OrienterNet accommodates different camera characteristics (e.g. focal lengths and image size) and poses information, thereby leveraging the full information available from Mapillary.

To incorporate pose information, we adopt OrienterNet's approach of gravity-aligning the image, ensuring that each column aligns with a gravity-aligned vertical plane in 3D [33]. Gravity alignment is essential for matching with OpenStreetMap data, as many Mapillary images are captured from vehicles on slopes, or from bicycles and hand-held cameras with varying tilt. After gravity alignment, the image is resized and padded to 512 x 512 pixels.

The processed, gravity-aligned image is then passed through an image encoder to obtain first-person view (FPV) features. Unlike OrienterNet [33], we replace its ResNet encoder with a DINOv2 ViT-

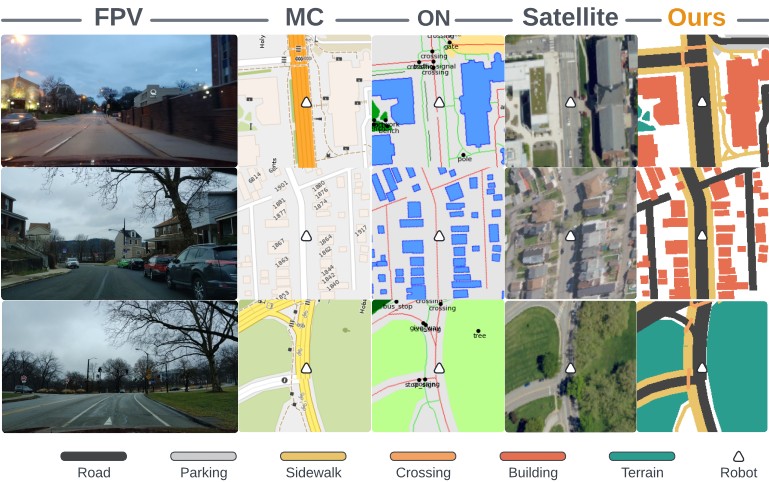

Figure 11: Qualitative comparisons between different rendering/rasterizing pipelines. ON: OrienterNet[33]. MIA's rendering pipeline produces semantic map suitable for map prediction.

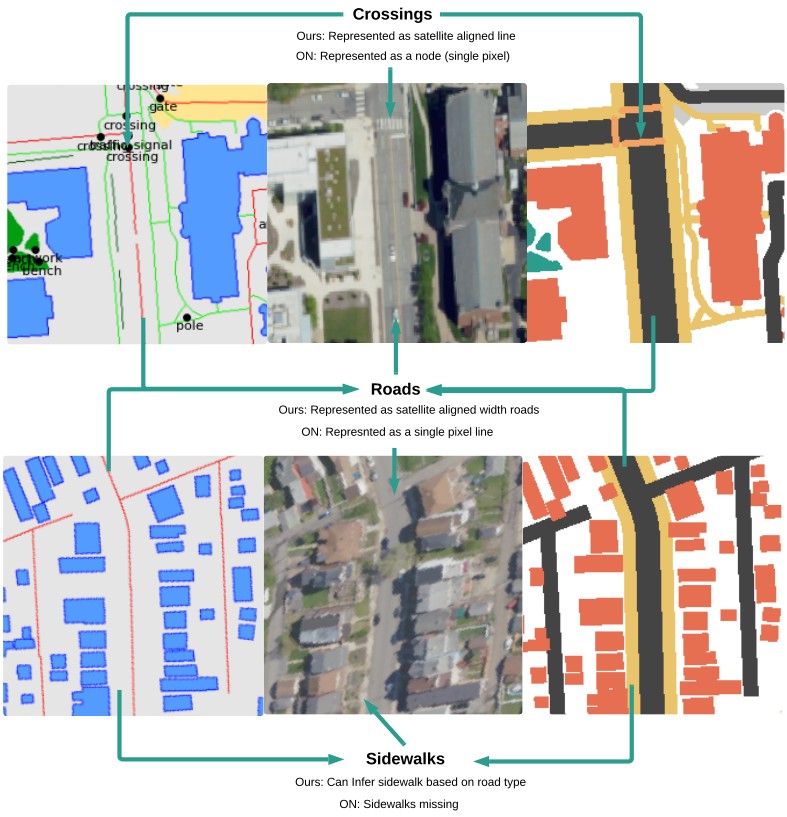

Figure 12: Detailed comparison of BEV rendering/rasterizing pipelines between OrienterNet[33] (left) and Ours (Right). Our road widths are more satellite aligned than OrienterNet's roads which are single pixel lines. Our crossings are drawn out from sidewalk to sidewalk whereas OrienterNet crossings are single pixel nodes. We are able to infer sidewalks based on road type in accordance with the OSM standards whereas OrienterNet does not.

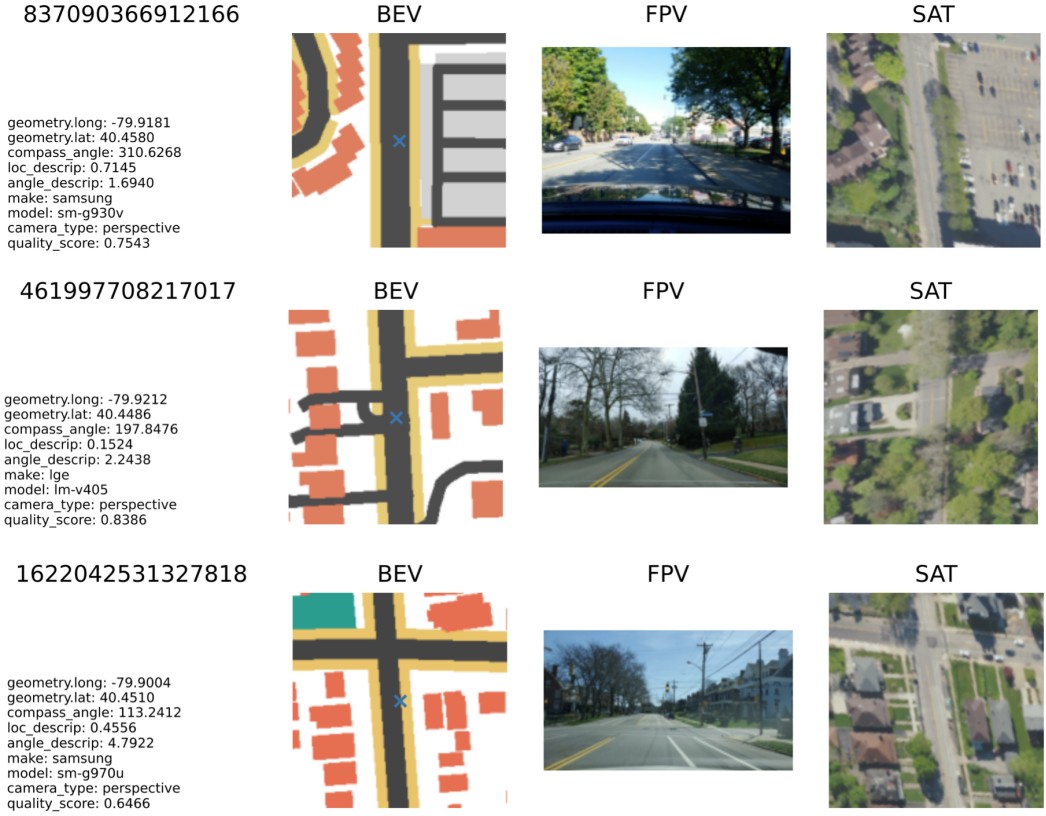

Figure 13: Additional `MIA` data samples and associated metadata

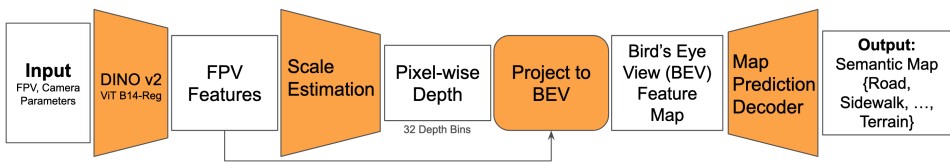

Figure 14: `Mapper` **Architecture**

B/14 [28] with registers [10] as the image encoder for improved generalizability. The FPV features are then fed into a linear layer to estimate pixel-wise scores for one of 32 *scale* (depth normalized by focal length) bins. These scale scores are then mapped to metric depth scores based on the focal length. The depth scores are used to project FPV features to generate a bird's eye view feature map through polar and Cartesian projections. To adapt OrienterNet [33] to the map prediction task, we add a decoder head to the BEV features to predict the semantic map.

## A.7  Hyperparameters

We release our pipeline hyperparameters in this section. We use Table 7 in our data engine; Table 8 for pretraining `Mapper` using `MIA` dataset; and Table 9, Table 10 for NuScenes and KITTI-360 fine-tuning.

## A.8  Dataset Privacy and Consent

Our First-Person-View data source, Mapillary [7], employs measures to ensure privacy by blurring faces and license plates, thereby removing personally identifiable information. Detailed information on their privacy policy is available here. When users contribute images to Mapillary, these images are shared under the CC-BY-SA license. Further details on this licensing can be found here.

Table 7: Data Curation Hyperparameters

| Parameter | Value |
|---|---|
| $\alpha$ | 224 |
| $\delta$ | 50 |
| $\rho$ | 0.5 |
| Recency Filter | > 2017 |
| Location Discrepancy | 3m |
| Angle Discrepancy | 20° |
| Sparsity Filter | 4m |
| Camera Models | "hdr-as200v", "iphone11pro", "iphone11", "iphone12", "gopromax", "iphone12pro", "lm-v405", "iphone11promax", "hdr-as300", "iphone13", "fdr-x1000v", "sm-g970u", "sm-g930v", "iphone13promax", "iphone13pro", "iphone12promax", "fdr-x3000" |

Table 8: Pretraining Hyperparameters

| Data | | Model | |
|---|---|---|---|
| **Parameter** | **Value** | **Parameter** | **Value** |
| Resize Image | 512 | Backbone Model | DINOv2 ViT-B/14 w/ registers |
| Batch Size | 128 | Latent Dimension | 128 |
| Gravity Alignment | Yes | Dropout Rate | 0.2 |
| Pad To Square | Yes | Learning Rate | 1.00E-03 |
| Rectify Pitch | Yes | LR Scheduler | Cosine Annealing LR |
| Scenes | Chicago, New York, Los Angeles, San Francisco | Losses | Dice Loss + Binary Cross Entropy Loss |
| Augmentations | Brightness, Contrast, Saturation, Random Flip, Hue | Loss Mask | Frustum |
| | | Class Weights | [1.00351229, 4.34782609, 1.00110121, 1.03124678, 6.69792364, 7.55857899] |

Table 9: Finetuning (NuScenes) Hyperparameters

| Data | | Model | |
|---|---|---|---|
| **Parameter** | **Value** | **Parameter** | **Value** |
| Resize Image | 512 | Backbone Model | DINOv2 ViT-B/14 w/ registers |
| Batch Size | 128 | Latent Dimension | 128 |
| Gravity Alignment | Yes | Dropout Rate | 0 |
| Pad To Square | Yes | Learning Rate | 1.00E-04 |
| Rectify Pitch | Yes | LR Scheduler | Cosine Annealing LR |
| Scenes | All (Only Front Camera) | Losses | Dice Loss + Binary Cross Entropy Loss |
| Augmentations | Brightness, Contrast, Saturation, Random Flip, Hue | Loss Mask | Frustum |
| | | Class Weights | [1.00060036, 1.85908161, 1.0249052, 2.57267816] |

Our Bird's Eye Map (BEV) data source, OpenStreetMap [9] provides guidance to limit mapping private information. Details can be found here.

More licensing information can be found in Table 11.

Table 10: Finetuning (KITTI360-BEV) Hyperparameters

| Data | | Model | |
|---|---|---|---|
| **Parameter** | **Value** | **Parameter** | **Value** |
| Target Focal Length | 256 | Backbone Model | DINOv2 ViT-B/14 w/ registers |
| Batch Size | 32 | Latent Dimension | 128 |
| Gravity Alignment | No | Dropout Rate | 0.1 |
| Pad To Square | Yes | Learning Rate | 1.00E-04 |
| Rectify Pitch | Yes | LR Scheduler | Cosine Annealing LR |
| Scenes | All (Only Front Camera) | Losses | Dice Loss + Binary Cross Entropy Loss |
| Augmentations | Brightness, Contrast, Saturation, Random Flip, Hue | Loss Mask | Frustum + Visibility |
| | | Class Weights | [2.5539, 3.8968, 1.9405, 5.6612] |

Table 11: Licenses for all projects and datasets used throughout this work. "*" denotes works used for the development of MIA. "†" denotes works used for evaluation purposes only.

| Projects | License |
|---|---|
| Mapillary* [7] | CC BY-SA 4.0 |
| Open Street Map* [9] | ODbL |
| MapMachine* [37] | MIT License |
| Nuscenes† [4] | CC BY-NC-SA 4.0 |
| KITTI-360† [23] | CC BY-NC-SA 3.0 |
| KITTI-360-BEV† [15] | Non Commercial Use Only |
| OrienterNet* [33] | CC-BY-SA 4.0 |

