# 1 Hosting, Licensing, and Maintenance Plan

Our dataset, `MIA`, will be hosted by AirLab at Carnegie Mellon University (CMU). The dataset will be available for a minimum of five years, with no plans for removal. We will ensure ongoing maintenance to verify and maintain data accessibility. The dataset can be accessed via https://github.com/MapItAnywhere/MapItAnywhere/blob/main/mia/dataset.md. The first-person-view images and associated metadata of the `MIA` dataset are published under the CC-By-SA license, similar to Mapillary. The bird's eye view map of `MIA` dataset is published under ODbL following OpenStreetMap. **We, the authors, bear all responsibility in case of violation of rights, etc., and confirmation of data license.**

# 2 Datasheet

## 2.1 Motivation

1. **For what purpose was the dataset created? Was there a specific task in mind? Was there a specific gap that needed to be filled? Please provide a description.**
   The `MIA` data engine and dataset were created to accelerate research progress towards *anywhere* map prediction. Current map prediction research builds on only a few datasets released by autonomous vehicle companies, which cover very limited areas. We therefore present the `MIA` data engine, a more scalable approach that uses large-scale crowd-sourced mapping platforms, Mapillary for FPV images and OpenStreetMap for BEV semantic maps. We show `MIA`'s utility by sampling the `MIA` dataset which spans 6 cities, and, to the best of our knowledge, is 6x more coverage than the closest publicly available dataset.

2. **Who created the dataset (e.g., which team, research group) and on behalf of which entity (e.g., company, institution, organization)?**
   The dataset is curated by AirLab at CMU.

3. **Who funded the creation of the dataset? If there is an associated grant, please provide the name of the grantor and the grant name and number.**
   The dataset curation is funded by National Institute of Advanced Industrial Science and Technology (AIST), Army Research Lab (ARL) awards W911NF2320007 and W911NF1820218 and W911NF20S0005, Defence Science and Technology Agency (DSTA). Omar Alama is partially funded by King Abdulaziz University.

4. **Any other comments?**
   No

## 2.2 Composition

5. **What do the instances that comprise the dataset represent (e.g., documents, photos, people, countries)?**
   We provide data needed for Bird's Eye View map prediction from a first-person image of a place under different environmental conditions and from different geographies. We curate data from 6 urban cities: New York, Chicago, Houston, Los Angeles, Pittsburgh, and San Francisco, covering approximately 470 $km^2$.

6. **How many instances are there in total (of each type,if appropriate)?** We provide 1,204,107 samples for `MIA`. Each sample includes: 1 First-Person-View image, 1 metadata file about the FPV RGB image, 1 associated top-down semantic mask, and 1 associated top-down visibility mask.

7. **Does the dataset contain all possible instances or is it a sample (not necessarily random) of instances from a larger set?**
   The dataset samples instances from a larger set from Mapillary and OpenStreetMap that our `MIA` data engine can obtain. As described in the paper, we chose to sample six different urban-centered U.S. cities to show `MIA`'s utility as a map prediction dataset. We selected highly populated cities - New York, Chicago, Houston, and Los Angeles - to collect challenging scenarios with diverse and dense traffic. Additionally, we included Pittsburgh and San Francisco for their unique topologies. Within the city boundaries we mined from the sample, the dataset is representative in terms of geographic coverage with respect to the FPV images

that meets our filtering mechanisms described in Section 3 (e.g., removing images with high pose noise, etc.). As mentioned in the paper, we currently only collect from 6 cities in the US, although it is possible to mine worldwide through our data engine.

8. **What data does each instance consist of? "Raw" data(e.g., unprocessed text or images) or features? In either case, please provide a description.**

For the FPV portion, each sample contains a FPV image and camera pose metadata (needed for BEV transformations) which we curate from Mapillary. Metadata fields are: 'id', 'height', 'width', 'camera_parameters', 'camera_type', 'captured_at', 'compass_angle', 'altitude', 'computed_compass_angle', 'computed_altitude', 'computed_rotation', 'thumb_2048_url', 'thumb_original_url', 'sequence', 'make', 'model', 'is_pano', 'quality_score', 'exif_orientation', 'geometry.coordinates', 'computed_geometry.coordinates', 'sfm_cluster.id', 'sfm_cluster.url', 'creator.username', 'creator.id', 'geometry.long', 'geometry.lat', 'computed_geometry.long', 'computed_geometry.lat'. The metadata fields are derived from Mapillary following convention used in OrienterNet [1]. For more information about the fields, please refer to Mapillary API Documentation. While we provide both `geometry` (pose directly from GPS) and `computed_geometry` (pose from Mapillary after processing with Structure from Motion) fields, we use the `computed_geometry` in the `MIA` data engine to pair the semantic mask.

For the BEV portion, each sample contains a semantic mask where the camera position is at the bottom-center of the mask. Each pixel represents 0.5 meters. The mask is $224 \times 224 \times 8$ in size, where the last dimension maps to classes as follows {`crossing`, `explicit_pedestrian`, `park`, `building`, `water`, `terrain`, `parking`, `train`} In addition, we provide a $224 \times 224$ top-down binary observable region mask that was obtained with a heuristic-based raycast function described in Section 5.

9. **Is there a label or target associated with each instance? If so, please provide a description.**
The semantic mask is the label associated with each instance. The semantic mask is a numpy file where each layer in the array is a binary mask for a given class.

10. **Is any information missing from individual instances? If so, please provide a description, explaining why this information is missing (e.g., because it was unavailable). This does not include intentionally removed information, but might include, e.g., redacted text.**
No.

11. **Are relationships between individual instances made explicit (e.g., users' movie ratings, social network links)? If so, please describe how these relationships are made explicit.**
Relationships between FPV image and BEV semantic mask is made explicit by its ID.

12. **Are there recommended data splits (e.g., training, development/validation, testing)? If so, please provide a description of these splits, explaining the rationale behind them.**
We include information regarding the data splits in the main paper and appendix. The train, val and testing data splits that we provide (named `split.json`) are geographically separated. This enables better testing of generalizable map prediction research.

13. **Are there any errors, sources of noise, or redundancies in the dataset? If so, please provide a description.**
The camera may have pose error due to GPS noise, especially in urban canyons. To minimize such errors, in the `MIA` data engine pipeline, we filter out images with a high pose discrepancy between the pose given by the camera's GPS (`geometry`) the pose calculated by Mapillary with Structure from Motion (`computed_geometry`). More information on geometry metadata fields can be found in Mapillary API documentation.However, despite such mechanism, pose error will still be present. As the Bird's Eye View map data is from crowdsourced data, some labeling errors are unavoidable. Examples include mislabeling of sidewalks, or mislabeling of low-frequency classes. In addition, we also find label errors in settings where multiple entities exist along a vertical column (e.g. bridges), however, such scenarios are rare in our dataset. While such label errors exist, we found through experimentation that the model trained with the dataset exhibits better generalization performance compared to baseline models trained with conventional autonomous vehicle

datasets. There are no redundancies in the data, as it represents different instances of camera images (time and location).

14. **Is the dataset self-contained, or does it link to or otherwise rely on external resources (e.g., websites, tweets, other datasets)?**
Yes, the dataset is self-contained.

15. **Does the dataset contain data that might be considered confidential (e.g., data that is protected by legal privilege or by doctor–patient confidentiality, data that includes the content of individuals' non-public communications)? If so, please provide a description.**
Our dataset is sourced from two platforms: Mapillary and OpenStreetMap, each addressing privacy concerns through its own policy. Mapillary's privacy policy is found https://www.mapillary.com/privacy, which describes their use of technology to blur faces and license plates to help reduce the impact on privacy. Additionally, Mapillary's terms require users to report any imagery that may contain personal data. OpenStreetMap imposes restrictions on mapping private information that "violates the privacy of people living in this world," with guidelines found here.

16. **Does the dataset contain data that, if viewed directly, might be offensive, insulting, threatening, or might otherwise cause anxiety? If so, please describe why.**
No

17. **Does the dataset relate to people? If not, you may skip the remaining questions in this section.**
No

18. **Does the dataset identify any subpopulations(e.g., by age, gender)?**
No

19. **Is it possible to identify individuals (i.e., one or more natural persons), either directly or indirectly (i.e., in combination with other data) from the dataset? If so, please describe how.**
No

20. **Does the dataset contain data that might be considered sensitive in any way (e.g., data that reveals racial or ethnic origins, sexual orientations, religious beliefs, political opinions or union memberships, or locations; financial or health data; biometric or genetic data; forms of government identification, such as social security numbers; criminal history)? If so, please provide a description.**
No

21. **Any other comments?**
No

## 2.3 Collection Process

22. **How was the data associated with each instance acquired?**
The paired FPV and BEV semantic data is acquired, associated and validated using the `MIA` data engine described in the main paper.

23. **What mechanisms or procedures were used to collect the data (e.g., hardware apparatus or sensor, manual human curation, software program, software API)? How were these mechanisms or procedures validated?**
We use the `MIA` data engine on our CPU cluster to obtain the data. We manually inspect to validate a sample of the dataset.

24. **If the dataset is a sample from a larger set, what was the sampling strategy (e.g., deterministic, probabilistic with specific sampling probabilities)?**
The dataset is a sample of areas that `MIA` can curate data from. The samples are chosen deterministically given our location parameters.

25. **Who was involved in the data collection process (e.g., students, crowdworkers, contractors) and how were they compensated (e.g., how much were crowdworkers paid)?**
We curate `MIA` dataset from data already available on two open-source mapping platforms: Mapillary and OpenStreetMap. The automatic curation of the dataset was run and validated by the authors, where the funding sources for the different authors have been acknowledged.

26. **Over what timeframe was the data collected? Does this timeframe match the creation timeframe of the data associated with the instances (e.g., recent crawl of old news articles)? If not, please describe the time- frame in which the data associated with the instances was created.**
We curate First-Person-View data that is taken after 2017, as described by Mapillary. For BEV map, we curate the latest available information from OpenStreetMap.

27. **Were any ethical review processes conducted (e.g., by an institutional review board)? If so, please provide a description of these review processes, including the outcomes, as well as a link or other access point to any supporting documentation.**
No

28. **Did you collect the data from the individuals in question directly, or obtain it via third parties or other sources (e.g., websites)?**
N/A

29. **Were the individuals in question notified about the data collection? If so, please describe (or show with screenshots or other information) how notice was provided, and provide a link or other access point to, or otherwise reproduce, the exact language of the notification itself.**
N/A

30. **Did the individuals in question consent to the collection and use of their data? If so, please describe (or show with screenshots or other information) how consent was requested and provided, and provide a link or other access point to, or otherwise reproduce, the exact language to which the individuals consented.**
N/A

31. **If consent was obtained, were the consenting individuals provided with a mechanism to revoke their consent in the future or for certain uses? If so, please provide a description, as well as a link or other access point to the mechanism (if appropriate).**
N/A

32. **Has an analysis of the potential impact of the dataset and its use on data subjects (e.g., a data protection impact analysis) been conducted?**
N/A

33. **Any other comments?** No

## 2.4 Preprocessing/cleaning/labeling

34. **Was any preprocessing/cleaning/labeling of the data done (e.g., discretization or bucket- ing, tokenization, part-of-speech tagging, SIFT feature extraction, removal of instances, processing of miss- ing values)? If so, please provide a description. If not, you may skip the remaining questions in this section.**
We curate the FPV and BEV data pairs with the `MIA` data engine described in Section **??** of the main paper.

35. **Was the "raw" data saved in addition to the preprocessed/cleaned/labeled data (e.g., to support unanticipated future uses)? If so, please provide a link or other access point to the "raw" data.**
No

36. **Is the software that was used to preprocess/clean/label the data available? If so, please provide a link or other access point.**
Yes, we open source the code for the `MIA` data engine at the following link: https://github.com/MapItAnywhere/MapItAnywhere.

37. **Any other comments?**
No

## 2.5 Uses

38. **Has the dataset been used for any tasks already? If so, please provide a description.**
The dataset was used for the map prediction task in this paper. The goal of map prediction is to predict a Bird's Eye View semantic map from First-Person-View images.

39. **Is there a repository that links to any or all papers or systems that use the dataset? If so, please provide a link or other access point.**
No. We will maintain a list of paper that uses our dataset in our GitHub repository. https://github.com/MapItAnywhere/MapItAnywhere.

40. **What (other) tasks could the dataset be used for?** The dataset can be used for various top-down map related tasks such as localization and planning.

41. **Is there anything about the composition of the dataset or the way it was collected and preprocessed/cleaned/labeled that might impact future uses?**
No

42. **Are there tasks for which the dataset should not be used? If so, please provide a description.**
No

## 2.6 Distribution

43. **Will the dataset be distributed to third parties outside of the entity (e.g., company, institution, organization) on behalf of which the dataset was created? If so, please provide a description.** Links to dataset is available at https://github.com/MapItAnywhere/MapItAnywhere/blob/main/mia/dataset.md.

44. **How will the dataset will be distributed (e.g., tarball on website, API, GitHub)? Does the dataset have a digital object identifier (DOI)?**
The dataset is distributed via a publicly accessible cloud storage, the links to the dataset can be found at https://github.com/MapItAnywhere/MapItAnywhere/blob/main/mia/dataset.md. MIA data engine code is at https://github.com/MapItAnywhere/MapItAnywhere. The DOI for dataset and repository is 10.5281/zenodo.11636653.

45. **When will the dataset be distributed?** Links to the presently-available dataset can be found at https://github.com/MapItAnywhere/MapItAnywhere/blob/main/mia/dataset.md.

46. **Will the dataset be distributed under a copyright or other intellectual property (IP) license, and/or under applicable terms of use (ToU)? If so, please describe this license and/or ToU, and provide a link or other access point to, or otherwise reproduce, any relevant licensing terms or ToU, as well as any fees associated with these restrictions.**
The FPV images and its associated metadata of MIA dataset is published under CC-By-SA following Mapillary. The BEV components of MIA dataset is published under ODbL following OpenStreetMap.

47. **Have any third parties imposed IP-based or other restrictions on the data associated with the instances? If so, please describe these restrictions, and provide a link or other access point to, or otherwise reproduce, any relevant licensing terms, as well as any fees associated with these restrictions.**
No. Mapillary images are available under CC-By-SA. OpenStreetMap data is available under ODbL.

48. **Do any export controls or other regulatory restrictions apply to the dataset or to individual instances? If so, please describe these restrictions, and provide a link or other access point to, or otherwise reproduce, any supporting documentation.**
No

49. **Any other comments?**
No

## 2.7 Maintenance

50. **Who will be supporting/hosting/maintaining the dataset?**
AirLab at CMU will support, host and maintain the dataset.

51. **How can the owner/curator/manager of the dataset be contacted (e.g., email address)?**
The manager can be contacted via our GitHub issues of the GitHub repository at https://github.com/MapItAnywhere/MapItAnywhere.

52. **Is there an erratum? If so, please provide a link or other access point.**
There is no erratum for current release. Future erratums, if applicable, will be available at https://github.com/MapItAnywhere/MapItAnywhere/blob/main/mia/dataset.md.

53. **Will the dataset be updated (e.g., to correct labeling errors, add new instances, delete instances)? If so, please describe how often, by whom, and how updates will be communicated to dataset consumers (e.g., mailing list, GitHub)?**
We plan to add new modalities and new locations in future dataset iterations. Updates will be communicated on the GitHub Repository at https://github.com/MapItAnywhere/MapItAnywhere.

54. **If the dataset relates to people, are there applicable limits on the retention of the data associated with the instances (e.g., were the individuals in question told that their data would be retained for a fixed period of time and then deleted)? If so, please describe these limits and explain how they will be enforced.**
No

55. **Will older versions of the dataset continue to be supported/hosted/maintained? If so, please describe how. If not, please describe how its obsolescence will be communicated to dataset consumers.**
Yes, we aim to continue supporting the `MIA` dataset.

56. **If others want to extend/augment/build on/contribute to the dataset, is there a mechanism for them to do so? If so, please provide a description. Will these contributions be validated/verified? If so, please describe how. If not, why not? Is there a process for communicating/distributing these contributions to dataset consumers? If so, please provide a description.**
Yes, contributors can propose changes via our GitHub issues in our repository at https://github.com/MapItAnywhere/MapItAnywhere.

57. **Any other comments?**
No

# References

[1] Paul-Edouard Sarlin, Daniel DeTone, Tsun-Yi Yang, Armen Avetisyan, Julian Straub, Tomasz Malisiewicz, Samuel Rota Bulo, Richard Newcombe, Peter Kontschieder, and Vasileios Balntas. OrienterNet: Visual Localization in 2D Public Maps with Neural Matching. In *CVPR*, 2023. 2