# OpenReview forum: "Map It Anywhere: Empowering BEV Map Prediction using Large-scale Public Datasets"
_NeurIPS.cc/2024/Datasets_and_Benchmarks_Track — NeurIPS 2024 Track Datasets and Benchmarks Poster_

### Official Review · Reviewer_Eqmy · 2024-07-25
**Automatic generation of good dataset**

**Rating:** 7
**Confidence:** 3
**Correctness:** The paper seems to be correct.
**Clarity:** The writing is clear and easy to follow.

**Review:**

+ Pipeline to automate the dataset creation process.
+ Evaluation on the dataset show performance improvement.

- Lack some diverse locations in other countries.
- Lack some detail in comparison with OrienterNet.

The writing is good and clear to follow. This work is build upon OrienterNet, which has a different map notion and not as suitable for BEV prediction. However, a more detailed comparison would be beneficial to highlight the difference between this work and OrienterNet, which may have a broader audience and application than this work. The scope of this work is focused on BEV prediction from FPV image. While the authors claim the importance of the BEV prediction for robotics application, the actual importance and broader impact is not very clear in the paper.

**Strengths:**

The strength of this submission is the pipeline to automatically generate the BEV prediction dataset based on crowd sourced data. The pipeline can potentially be adapted to other applications.

**Additional Feedback:**

None

**Documentation:**

The documentation is adequate.

**Ethics:**

No ethical concerns.

**Limitations:**

The included discussion on limitations and societal impact is adequate. But some limitations seems should be solved in this work (locations outside of US). Privacy risk of crowd sourced data is not addressed.

**Opportunities For Improvement:**

The work should include locations outside of the US, such as some places in the EU and Asia for diversity and robustness of the pipeline. This seems to be should be an easy extension and included in this submission to support the claim that MIA data engine is applicable to other world-wide locations.

**Relation To Prior Work:**

The relation to prior work OrienterNet is not very clear and a more detailed discussion would be appreciated.

**Summary And Contributions:**

The authors propose a new way to generate bird's eye view maps that can be used to develop methods to predict BEV map from first person view images. The contribution is the pipeline to automate the process and potentially create an ever increasing dataset.

---

> ### Author Rebuttal · Authors · 2024-08-17
>
> **Strengths**
> Thank you for your insightful feedback\! We are delighted to hear you are excited about our work which introduces an automatic pipeline to build an ever-increasing BEV prediction dataset. We are excited you think it can be potentially adapted to other applications, and that the experiments show improved results.
>
> We have put significant thought into your feedback and have curated a new dataset from non-US cities and conducted experiments to address your concerns\!
>
> **Clarify the difference between our work and OrienterNet \[26\]**
> Thank you for bringing up that we should make the differences between OrienterNet \[26\] and MIA clearer. To clarify, MIA builds on OrienterNet which uses Mapillary and OpenStreetMap, to perform RGB image localization within a pre-existing map. Our work extends OrienterNet by using the same data sources to achieve the goal of generalizable FPV-to-BEV map prediction. As stated in the global rebuttal, our work differs from OrienterNet in 2 ways: dataset processing pipeline and released datasets. Below we summarize the differences for your convenience:
>
> * **Data processing pipeline:** we stated the differences in lines 93-99 of the submission. First, OrienterNet has a fixed set of image IDs that they have curated either manually or through a closed-source method. In contrast, we developed a filtering pipeline that allows the automatic curation and collection of images from any location in the world, where users can define a city of choice and change filtering thresholds to meet their needs. Secondly, OrienterNet-rendered maps, while useful to give context to an image encoder, are unsuitable as a prediction target and hence unsuitable for BEV prediction. This is because the map elements are not consistent with the real world / FPV image. For example, roads in OrienterNet are 1 pixel wide, whereas the road in ours is a rasterized shape that is aligned with satellite and FPV images. We illustrate road, crossing, and sidewalk differences in Fig. 3 & 4\.
> * **Released datasets:** OrienterNet released the MGL dataset. As stated in our submission's Tab. 1,  MGL maps are unsuitable for BEV prediction. Whereas, our dataset is not only suitable for BEV map prediction but is also curated automatically with only city names as input. For more clarity, we changed the row name to "OrienterNet (MGL)", and added a column for "suitable for semantic map prediction", where OrienterNet (MGL) is marked as unsuitable, while MIA is marked as suitable.
>
> We believe MIA opens both works up to a new set of audiences interested in operating without a pre-existing map. As it is mainly a localization module, deploying OrienterNet will require access to an existing map. However, there is a strong assumption that the map is accurate for downstream tasks. While OpenStreetMap has been sufficient for our training purposes, solely relying on it may be detrimental in case of erroneous labels and map updates. In contrast, our map predictor infers the map given a front-view image and IMU orientation.
>
> **Highlight the importance and broader impact of BEV prediction:**
> We agree with the reviewer that the actual importance of BEV prediction can be better highlighted\!
> We believe MIA's BEV mapping technique will significantly improve ground robot navigation performance in outdoor environments due to its improved generalizability.
> The output metric semantic map can be used by the robot to define where is desirable to traverse and plan paths given its capabilities, for example, an autonomous wheelchair should stay on the sidewalk.
> A key advantage of MIA is that we can curate data from a much more diverse range of scenarios, e.g. pedestrian sidewalks, and park footpaths, than conventional autonomous driving datasets which are all taken from cars on roads, thereby allowing more types of ground robots to use this map predictor. Beyond robot navigation, while we don't focus on these applications, our BEV maps can be extended for cross-view localization similar to \[26, Reb6\] and building global maps similar to \[Reb7\].  For more information, refer to the BEV mapping section of the global rebuttal.
>
> **Including non-US locations, to show the diversity and robustness of the pipeline**
> Thank you for highlighting the need to include non-US locations\! MIA’s key strength is its ability to *automatically* curate data from data sources across the world. While MIA has shown promising results in US cities, we agree that demonstrating its ability to curate data and predict BEV maps in non-US cities will help highlight the value of our data-curation pipeline.  To this end, based on your feedback, we made the following changes:
>
> * We extracted a dataset of 284K images from 6 non-US cities spanning 3 continents: Europe (London, Stockholm, Berlin, Zurich), Asia (Tokyo), and Australia (Sydney) \[Rebuttal Fig.1\].  The data is uploaded [here](http://cmu.box.com/s/q24osfhfvypqwnmnaaszozbrh8pfluc7).
> * We evaluated MIA zero-shot on the above dataset. Results in Tab. 1 and Fig. 2 show that our model Mapper consistently provides accurate and realistic zero-shot predictions in comparison to baselines trained on conventional datasets.
> * We built an [online demo](http://bit.ly/3SMLZqv) to showcase our data engine's ability to sample {image, BEV semantic data} pairs from a user's choice of cities.  We encourage the reviewers to try with a city of their choice.
>
> **Addressing Privacy Risk of Crowdsourced data**
> Thank you for bringing up the important topic of privacy risk. We kindly refer you to see lines 494-499 on how our data sources handle dataset privacy. Mapillary blurs face and license plates to remove personally identifiable information. More information can be found in their [privacy policy](http://www.mapillary.com/privacy). OpenStreetMap provides guidance and rules to limit mapping private information. More information can be found on their [website](http://shorturl.at/slLIF).

---

### Official Review · Reviewer_NDfz · 2024-07-25

**Rating:** 6
**Confidence:** 3

**Review:**

### Pros

- Large-Scale Dataset Creation: MIA facilitates the automatic collection of a significant dataset, comprising 1.2 million pairs of First-Person View (FPV) and BEV images across diverse geographical contexts.
- Improved Generalization and Robustness: The methodology demonstrates marked improvements in BEV map prediction performance compared to prior approaches, indicating greater generalizability across varied environments.

### Cons

* The reliance on crowd-sourced data has inherent limitations, including label noise and potential bias, particularly since the dataset is skewed towards locations in the US. I suggest the authors conduct further generalization experiments on other locations beyond the US.
* The approach primarily focuses on static maps, which means it may not adequately capture dynamic classes, limiting its applicability in environments with moving objects (like pedestrians or vehicles), while these elements are critical in real-world autonomous navigation applications.

**Strengths:**

* **Large-Scale Dataset Collection**: This dataset captures diverse geographical contexts, significantly enhancing the robustness and adaptability of BEV mapping techniques.
* **Establishing a Benchmark**: MIA not only provides a vast dataset but also establishes a benchmark for future research in BEV mapping.

**Additional Feedback:**

NA

**Clarity:**

* This paper is well-organized and easy to follow.

**Correctness:**

* The paper accurately reflects its contributions and scope. It properly addresses a important limitation in the urban research.

**Documentation:**

* The documentation is complete and well-organized.

**Limitations:**

* See Cons

**Opportunities For Improvement:**

* See Cons

**Relation To Prior Work:**

* MIA addresses the limitation of previous work by utilizing large-scale public data from crowd-sourced platforms like Mapillary and OpenStreetMap, resulting in a more comprehensive dataset.
* MIA emphasizes improved generalizability. It shows that training a model using diverse data leads to superior zero-shot performance over state-of-the-art baselines.

**Summary And Contributions:**

This paper presents MIA, a framework designed to improve BEV mapping for ground robot navigation. Traditional BEV mapping methods often utilize limited datasets from autonomous vehicles, hindering their generalization across varied environments. MIA overcomes this limitation by automating the collection of a large dataset comprising 1.2 million pairs of FPV and BEV images, which capture a wide range of geographical contexts. Evaluations show that MIA significantly enhances BEV map prediction performance, surpassing previous SOTA approaches.

---

> ### Author Rebuttal · Authors · 2024-08-17
>
> **Strengths**
> Thank you for your insightful feedback\! We are delighted to hear that you believe that MIA addresses a key problem in generalizable BEV mapping. Thank you for highlighting that MIA "significantly enhances BEV map prediction performance, surpassing previous SOTA approaches".
>
> We have put significant thought into your feedback and have curated a new dataset from non-US cities and conducted experiments to address your concerns\!
>
> **Further generalization experiments on non-US locations:**
> Thank you for pointing out the skew towards locations in the US. We agree and have initially pointed this limitation out in lines 291-292. To address this limitation as per your suggestions, we have done the following:
>
> * We extracted a dataset of 284K images from 6 non-US cities spanning 3 continents: Europe (London, Stockholm, Berlin, Zurich), Asia (Tokyo), and Australia (Sydney) \[Rebuttal Fig.1\].  The data is uploaded [here](https://cmu.box.com/s/q24osfhfvypqwnmnaaszozbrh8pfluc7).
> * We evaluated MIA zero-shot on the above dataset. Results in Tab. 1 and Fig. 2 show that our model Mapper consistently provides more accurate and realistic zero-shot predictions in comparison to baselines trained on conventional datasets.
> * We built an [online demo](https://bit.ly/3SMLZqv) to showcase our data engine's ability to sample {image, BEV semantic data} pairs from a user's choice of cities.  We encourage the reviewer to try with a city of their choice.
>
> **Crowd-sourced Label Noise**
> Thank you for highlighting this limitation. Initially, we discussed this in lines 285-288 and provided more details in lines 116-130 in the supplementary. We agree that this is an important limitation of the work. Yet, we emphasize the following:
>
> * Large-scale / web-scale data have proven to be pivotal in the advancement of the generalizability of data-driven approaches despite varying degrees of (label) noise, such as ImageNet and Clothing1m. Our work recognizes this general trend and attempts to extend the approach of using large-scale data for generalization to FPV-BEV prediction.
> * Despite some label noise in our data, we show good performance using our model which was only trained on our dataset on conventionally collected and curated datasets such as Nuscenes and KITTI-360 compared with the baselines (please see Fig. 5 and Tab. 2 of our submission).
> * In addition, there is a body of work on detecting label noise and learning with noisy labels we can build on to further improve the quality of our dataset.
>
> **Handling dynamic classes:**
> Thank you for highlighting this limitation. We agree, and we initially acknowledged this in line 289\. While recognizing this, we want to highlight two points:
>
> * While MIA currently does not include dynamic objects,  we view our approach as complementary to autonomous vehicle (AV) datasets where dynamic classes exist. By combining the diversity of static maps provided by MIA with the dynamic labels from AV datasets, we believe we can achieve more generalizable static and dynamic class predictions. We are excited that together, MIA and conventional AV datasets can be used to pretrain for generalizable map prediction.
> * We further note that the prediction of the static elements is crucial for a robot to decide which path to take. While the labels do not include dynamic classes, the input images do include them in abundance allowing BEV prediction pipelines to reason beyond occluding dynamic objects.

---

> > ### Comment · Reviewer_NDfz · 2024-09-01
> > **acknowledgement**
> >
> > I acknowledge the author's response and efforts

---

### Official Review · Reviewer_GXHK · 2024-07-30

**Rating:** 6
**Confidence:** 4
**Clarity:** the language is good.

**Review:**

pros
+ creating a [RGB image - BEV map] dataset that encludes multiple camera models, time of day, weather, location
+ an automatic data collection process that can scale further
+ a strong baseline that can work with different camera parameters

cons
- it seems that [26] is a very similar work. from what the reviewer understands, [26] estimates the camera poses based on images and the rough GPS info, while this work should have access to more accurate camera pose from SfM and an online 'record'? the paper could use more clarification here.
- the manuscript needs more details on the raw data from Mapillary. what is the default format? what is included? from reading the paper the reviewer guesses that it should include both RGB images and their camera pose, but they cannot make sure of this from the Mapillary web or the paper.
- please provide more details on the geospatial filtering process. the angle discrepancy and the location discrepancy between SfM and recorded pose are used as cues. however, the reviewer cannot find details on the source of the SfM or the source of the camera pose. are they both from Mapillary? or are they computed by the authors?
- please provide more discussion on the potential use of this BEV mapping technique. perhaps include the retrieval of online maps, the cross-view localization, or something more.

summary
- overall, the reviewer likes this work and the potential it might have on building more generalizable solutions to BEV mapping. however, they feel that the current manuscript is somewhat lacking and could use more details and discussions to help the readers better understand what has been done before, what this work has done, and what can be done in the future.

**Strengths:**

see above

**Additional Feedback:**

n/a

**Correctness:**

the dataset seems to be well constructed. the documentation is not as detailed as the reviewer had hoped.

**Documentation:**

could use more details.

**Ethics:**

the authors have discussed the ethic issues.

**Limitations:**

see above

**Opportunities For Improvement:**

see above

**Relation To Prior Work:**

seems [26] is related but could use further clarification.

**Summary And Contributions:**

this paper focuses on bird's eye view (BEV) map estimation from perspective images and introduces a new framework for automatically collecting real-world data pairs of BEV maps and RGB images. by designing a complex filtering process and integrating two existing data source, open street map for BEV maps and Mapillary for RGB images, the authors successfully provided training data across multiple camera models, time of day, weather, and location. in addition, a baseline method, Mapper, is proposed which can take the RGB image and camera parameters (K and Rt) and estimate the BEV map. the proposed dataset enables strong zero-shot performance and can be used as pre-training data for downstream fine-tuning.

---

> ### Author Rebuttal · Authors · 2024-08-17
>
> **Strengths**
> Thank you for your insightful feedback\! We are delighted to hear that you liked our work and the potential for generalizable BEV mapping. We are excited that you highlighted that MIA enables strong zero-shot performance and can be used as pre-training data for downstream fine-tuning and provides a strong baseline that can work with different camera parameters.
>
> We have put significant thought into your feedback and have provided further clarifications below and made the following changes in response to them.
>
> **Expanding on differences with OrienterNet \[26\]**
> Thank you for your question clarifying the source of the camera pose. Both \[26\] and our work have access to the SfM pose provided by Mapillary API. Following \[26\], we use this SfM pose as ground-truth camera location.
>
> Thank you for your feedback on making the difference between Orienternet \[26\] and our work clearer. To clarify, MIA builds on OrienterNet \[26\] which uses Mapillary and OpenStreetMap, to perform RGB image localization within a pre-existing map. Our work extends OrienterNet by using the same data sources to achieve the goal of generalizable FPV-to-BEV map prediction. As stated in the global rebuttal, our work differs from OrienterNet in 2 ways: (1) dataset processing pipeline and (2) released datasets. Below we summarize the differences for your convenience:
>
> * **Data processing pipeline:** we stated the differences in lines 93-99 of the submission. First, OrienterNet \[26\] has a fixed set of image IDs that they have curated either manually or through a closed-source method. In contrast, we have developed a filtering pipeline that allows the automatic curation and collection of images from any location in the world, where users can define a city of choice and directly change desired filtering thresholds to meet their needs. Secondly, OrienterNet-rendered maps, while useful to give context to an image encoder, are unsuitable as a prediction target and hence unsuitable for BEV prediction. This is because the map elements are not consistent with the real world / FPV image. For example, roads in OrienterNet are 1 pixel wide, whereas the road in ours is a rasterized shape that is aligned with satellite and FPV images. We illustrate road, crossing, and sidewalk differences in the rebuttal's Fig. 3 & 4\.
> * **Released datasets:** OrienterNet released the MGL dataset. As stated in our submission's Tab. 1,  MGL maps are unsuitable for BEV prediction. Whereas, our dataset is not only suitable for BEV map prediction but is also curated automatically with only city names as input. For more clarity, we changed the row name to "OrienterNet (MGL)", and added a column for "suitable for semantic map prediction", where OrienterNet (MGL) is marked as unsuitable, while MIA is marked as suitable.
>
> **Mapillary raw data details:**
> We acknowledge that details on raw data from Mapillary are hard to find in our main paper or online. We kindly refer the reviewer to Lines 76-92 of our supplemental information where we detail the raw data (FPV image, camera pose, camera pose metadata) we curate from Mapillary. As you correctly pointed out, we obtain both the RGB images and camera poses from Mapillary, in addition to metadata about the image. Based on your suggestion, we have added to Line 124 of the original submission "For more information on the raw data we curate from Mapillary, please refer to Supplemental Information Q8." to better guide readers.
>
> **More details on Geospatial filtering:** We agree that lines 117-122 do not provide enough detail on this process. We revised those lines as follows:  “The FPV pipeline starts by manually inputting a list of locations of interest, which can be as simple as inputting the name of the location or as specific as specifying the GPS bounds. The geographical bounds are then fetched using the Nominatim API if needed. The pipeline then converts these bounds to a list of zoom-14 square tiles and uses the public Mapillary APIs to query for any image instances within these tiles. Only the retrieved instances that lie within the exact geographical boundaries of interest are kept.”
>
> **Clarification on SfM and pose sources:** We obtain both the SfM pose and camera pose from the Mapillary API. The camera pose is from the device sensors, which include GPS and IMU. The SfM rectified pose is computed by the Mapillary team running SfM on the image sequences. While this is stated in Line 91 of the supplementary, our main paper's line 128 does not make it clear where the source is from. We have added to line 128 the following to make it more clear: "a location/discrepancy filter that computes the difference between Structure from Motion (SfM) computed and recorded poses, **both obtained from Mapillary API,**  as a proxy for measuring the quality of the georegistration".
>
> **Highlighting use of BEV mapping technique**
> We believe MIA's BEV mapping technique will significantly improve ground robot navigation performance in outdoor environments due to its improved generalizability.
> The output metric semantic map can be used by the robot to define where is desirable to traverse and plan paths given its capabilities, for example, an autonomous wheelchair should stay on the sidewalk.
> A key advantage of MIA is that we can curate data from a much more diverse range of scenarios, e.g. pedestrian sidewalks, and park footpaths, than conventional autonomous driving datasets which are all taken from cars on roads, thereby allowing more types of ground robots to use this map predictor. Beyond robot navigation, while we don't focus on these applications, our BEV maps can be extended for cross-view localization similar to \[26, Reb6\] and building global maps similar to \[Reb7\].  For more information, please refer to the BEV mapping section of the global rebuttal.

---

### Author Rebuttal · Authors · 2024-08-17

We are excited to hear that our reviewers believe that MIA is a promising approach towards generalizable BEV mapping (GXHK, NDfz) and provides a large dataset (NDfz, Eqmy) for the map prediction task. We are glad that the reviewers found MIA to have strong map prediction performance, in zero-shot settings (GXHK) and compared to the SOTA (NDfz, Eqmy). We appreciate that the reviewers found our work to be readily extendible, for example, to scale data collection further (GXHK) and be adapted to other applications (Eqmy). Finally, we are happy the writing is clear and easy to follow (Eqmy).

We welcome reviewers to view our [website](https://mapitanywhere.github.io/), which contains demos of the data engine and our map predictor, and in-the-wild prediction examples.
Below, we summarize key areas of improvement highlighted by the reviewers and the changes and clarifications we have made in response to them.

**MIA's ability to curate and generalize to data from non-US cities (NDfz, Eqmy)**
MIA’s key strength is its ability to automatically curate data from data sources across the world. While MIA has shown promising results in US cities, we agree that demonstrating its ability to curate data and predict BEV maps in non-US cities will further highlight the value of our data-curation pipeline.  To this end, we made the following changes:

* We extracted a dataset of 284K images from 6 non-US cities spanning 3 continents: Europe (London, Stockholm, Berlin, Zurich), Asia (Tokyo), and Australia (Sydney) \[Rebuttal Fig.1\].  The data is uploaded [here](https://cmu.box.com/s/q24osfhfvypqwnmnaaszozbrh8pfluc7).
* We evaluated MIA in a zero-shot setting on the above dataset. Results in Tab. 1 and Fig. 2 show that our model Mapper consistently provides accurate and realistic zero-shot predictions compared to baselines trained on conventional datasets.
* We built an [online demo](http://bit.ly/3SMLZqv) to showcase our data engine's ability to sample {image, BEV semantic data} pairs from a user's chosen city.  We encourage the reviewers to try with a city of their choice.

**Differences between OrienterNet \[26\] and our work (GXHK, Eqmy)**
Thank you for highlighting the need to better clarify the differences between OrienterNet and MIA.
MIA builds on OrienterNet which uses Mapillary and OpenStreetMap, to perform RGB image localization within a pre-existing map. Our work extends OrienterNet by using the same data sources to achieve the goal of generalizable FPV-to-BEV map prediction. Our work differs in 2 ways: (1) dataset processing pipeline and (2) released datasets.
For the data processing pipeline, we stated the differences in lines 93-99 of the submission. First, OrienterNet has a fixed set of image IDs that was curated either manually or through a closed-source method. In contrast, we developed a filtering pipeline that allows the automatic curation and collection of images from any location in the world. Users can define a city and change filtering thresholds to meet their needs. Secondly, while OrienterNet-rendered maps provide useful context to an image encoder, they are unsuitable as a prediction target for BEV prediction. This is because the map elements are not consistent with the real world / FPV image. E.g., roads in OrienterNet are 1 pixel wide, whereas the road in ours is a rasterized shape aligned with satellite and FPV images. We illustrate the differences in the Fig. 3 & 4\.

Tab. 1 in our submission compares related datasets, including the MGL dataset released by OrienterNet. However, as stated in the caption, MGL maps are unsuitable for BEV prediction. Whereas, our dataset enables BEV prediction and was curated automatically with only city names as input.  For more clarity, we changed the row name to "OrienterNet (MGL)", and added a column for "suitable for semantic map prediction", where OrienterNet (MGL) is marked as unsuitable, while MIA is marked as suitable.

**Importance of Bird's Eye View (BEV) Mapping (GXHK, Eqmy)**
We thank the reviewers for pointing out the need to clarify the importance of BEV mapping. While we briefly stated the applications of bird's-eye views (BEV) mapping in Lines 23-24, we agree that its importance, and by extension, the significance of our work deserves more emphasis. BEV maps are an important environment representation for localization, mapping, and decision-making tasks, particularly for ground robots. They provide a rich, efficient metric representation of the world, enabling spatial reasoning directly in the top-down coordinate robots move in. This allows the robot to define in metric space how it should navigate around the world, i.e. head to and cross the crosswalk. The rich information supports nuanced decision-making, such as a delivery robot planning a path to stay on the sidewalk, instead of entering the road.
Given these advantages, autonomous systems often employ BEV maps as their primary representation, for example historically by the winner of DARPA Grand Challenge \[1\] and recently as the key representation of recent SOTA paper \[2\] on "Planning-Oriented Autonomous Driving". Beyond on-road driving, BEV mapping is widely used in other robot domains, such as offroad driving \[3\], mobile manipulation \[4\], and exploration \[5\]. In addition, BEV maps have been a popular representation for cross-view localization \[6\] and global mapping \[7\].
Such significance, coupled with the widespread use of cameras, has led to a growing body of work on FPV-to-BEV semantic mapping, as described in line 72 of our submission. However, these works rely on current autonomous driving datasets, which are limited in both scale and diversity due to their high cost. In contrast, MIA leverages data already available in crowdsourced platforms which contain data from across the world, dramatically increasing diversity and scale.
We have further highlighted the importance of BEV mapping in the revised manuscript.

---

> ### Author Response · Authors · 2024-08-22
> **Request for Feedback**
>
> Dear reviewers GXHK, NDfz, Eqmy,
>
> Thank you again for your valuable review. We have made several changes and clarifications based on your feedback, including a new dataset from 6 non-US cities from 3 continents, and more generalization experiments.
>
> Please let us know your thoughts. We are eager to improve our paper while we still have time.
>
> We understand you are busy and thank you for your time.
>
> Cheers,
>
> Authors

---

### Author Response · Authors · 2024-08-27
**Author/reviewer discussion period ending soon! Please let us know your thoughts.**

Dear reviewers GXHK, NDfz, Eqmy,

Thank you again for your time and valuable review. The author/reviewer discussion is ending soon!
We have made several changes based on your feedback, including a new dataset from 6 non-US cities from 3 continents, more generalization experiments, and clarifications. Please let us know your thoughts!

We understand you are busy and thank you for your time. We are looking forward to your response and are eager to improve our work.

Thank you!
Authors

---

### Decision · Program_Chairs · 2024-09-26

**Decision:**

Accept (Poster)

**Comment:**

This paper designs MIA, a framework aimed at enhancing BEV mapping for ground robot navigation. Traditional BEV mapping methods typically rely on limited datasets from autonomous vehicles, which restrict their ability to generalize across diverse environments. MIA addresses this limitation by automating the collection of an extensive dataset that includes 1.2 million pairs of FPV and BEV images, representing a broad spectrum of geographical contexts. I believe MIA can advance research in the field of Map prediction.

After reviewing the comments from the reviewers and the authors' feedback, it is found that the main questions and concerns have been addressed. The final ratings for this paper are **6, 6 and 7**.  These are consistently positive. I recommend accepting this paper and encourage the authors to improve the paper based on the suggestions of the reviewers.